# Heterogeneity of murine periosteum progenitors involved in fracture healing

Brya G Matthews[1,2]*, Sanja Novak[2], Francesca V Sbrana[2], Jessica L Funnell[2], Ye Cao[1], Emma J Buckels[1], Danka Grcevic[3,4], Ivo Kalajzic[2]

[1]Department of Molecular Medicine and Pathology, University of Auckland, Auckland, New Zealand; [2]Department of Reconstructive Sciences, UConn Health, Farmington, United States; [3]Department of Physiology and Immunology, University of Zagreb, Zagreb, Croatia; [4]Croatian Intitute for Brain Research, University of Zagreb, Zagreb, Croatia

**Abstract** The periosteum is the major source of cells involved in fracture healing. We sought to characterize progenitor cells and their contribution to bone fracture healing. The periosteum is highly enriched with progenitor cells, including Sca1[+] cells, fibroblast colony-forming units, and label-retaining cells compared to the endosteum and bone marrow. Using lineage tracing, we demonstrate that alpha smooth muscle actin ($\alpha$SMA) identifies long-term, slow-cycling, self-renewing osteochondroprogenitors in the adult periosteum that are functionally important for bone formation during fracture healing. In addition, Col2.3CreER-labeled osteoblast cells contribute around 10% of osteoblasts but no chondrocytes in fracture calluses. Most periosteal osteochondroprogenitors following fracture can be targeted by $\alpha$SMACreER. Previously identified skeletal stem cell populations were common in periosteum but contained high proportions of mature osteoblasts. We have demonstrated that the periosteum is highly enriched with skeletal progenitor cells, and there is heterogeneity in the populations of cells that contribute to mature lineages during periosteal fracture healing.

*For correspondence:
brya.matthews@auckland.ac.nz

Competing interests: The authors declare that no competing interests exist.

## Introduction

Bone tissue retains the ability to heal and regenerate throughout life. This process relies on tissue-resident stem and progenitor cells capable of generating new matrix. Until recently, most studies focused on the bone marrow compartment as a source of osteoprogenitors, with respect to their contribution to growth and remodeling, and their potential as a source of cells for regenerative applications. In the bone marrow compartment, it is well established that the endosteal region, as well as regions near trabecular bone surfaces, are highly enriched for stem and progenitor cells, including cells of the mesenchymal or skeletal lineage (*Siclari et al., 2013*; *Morikawa et al., 2009*). Skeletal stem cells (SSCs) can be isolated from various bone compartments based on their plastic adherence; however, more recently, numerous markers have been proposed to identify or isolate skeletal progenitors prospectively (*Cao et al., 2020*).

The skeleton has different requirements for the functionality of tissue-resident progenitors at different life stages. During development, the majority of the skeleton forms through endochondral ossification via a cartilage template, and this process continues during postnatal life at the growth plates. Hypertrophic chondrocytes can give rise to osteoblasts, particularly those in the primary spongiosa and trabecular region, likely via a progenitor intermediate (*Wolff and Hartmann, 2019*; *Yang et al., 2014a*; *Yang et al., 2014b*; *Zhou et al., 2014a*; *Mizuhashi et al., 2018*). However, this process is growth-related, and SSC pools must become established in adult tissues by the completion of growth when the growth plate activity reduces dramatically (as in mice) or fuses completely (in humans). Many stem and progenitor markers have been studied primarily during early life. For

example, skeletal progenitors expressing Gli1, Gremlin1, and CTGF are prevalent during development and adolescent growth but appear to diminish or disappear in mice over 1 month of age (*Shi et al., 2017*; *Worthley et al., 2015*; *Wang et al., 2015*). Chan and colleagues have reported flow cytometry-based methods for identifying SSCs and downstream bone, cartilage, and stromal progenitors (BCSPs), as well as various other lineage-restricted populations in mice and humans (*Chan et al., 2018*; *Chan et al., 2015*). The murine cell surface antigen combinations were validated only in cells isolated from whole neonatal mouse long bones, but have since been applied to various settings in adult systems, including fracture healing (*Marecic et al., 2015*; *Tevlin et al., 2017*). Similarly, markers for putative SSCs in humans were identified using fetal tissue, with a focus on the hypertrophic cartilage zone, then applied to adult tissue. Another recent study used similar marker combinations to identify stem cell populations in the periosteum, with the majority of studies performed using very young animals (late embryonic up to P32, many at P6) (*Debnath et al., 2018*). PDGFRα⁺Sca1⁺ (PαS) cells were characterized in adult animals, and leptin receptor Cre-labeled cells that ultimately give rise to osteoblasts and bone marrow adipocytes do not become established before 2 months of age (*Morikawa et al., 2009*; *Zhou et al., 2014b*). These results indicate that stem and progenitor cell populations and markers change at different life stages, meaning that markers identified in neonates or juveniles may not apply to the adult setting.

The periosteum is the tissue surrounding the outer surface of the bone. It is physically separated from the bone marrow compartment and appears to have a much greater regenerative capacity, playing a critical role in fracture healing (*Colnot, 2009*; *Roberts et al., 2015*). Periosteum-based healing involves a process similar to endochondral ossification. Following an inflammatory phase, the periosteum expands and forms a callus initially comprised of fibrocartilage in the central area, combined with direct bone formation at the periphery. It has been challenging to identify markers that are specific to the periosteum; however, recent studies have characterized cathepsin K and periostin as markers that are highly enriched in periosteal mesenchymal cells, and alpha smooth muscle actin (αSMA) was shown to enrich Mx1-labeled periosteal cells for progenitors (*Debnath et al., 2018*; *Duchamp de Lageneste et al., 2018*; *Ortinau et al., 2019*). We have previously shown that αSMA identifies a population of growth-related osteoprogenitors in the trabecular region, and cells that contribute to fracture healing in the adult periosteum (*Matthews et al., 2014*; *Grcevic et al., 2012*).

In the current study, we have used murine models to characterize the periosteal progenitor populations. We have demonstrated that αSMACreER identifies a subset of long-term osteochondroprogenitors involved in fracture callus formation. In addition, we have evaluated different bone compartments for the expression of stem/progenitor markers, including cell surface profile, SSC, and BCSP populations, and lifespan by label retention assays in different bone compartments, and evaluated growth potential of different periosteal populations ex vivo.

## Results

### In vivo identification of long-term osteochondroprogenitors

In order to identify murine periosteal progenitor populations in vivo, we initially utilized a lineage tracing approach with a tamoxifen-inducible Cre combined with the Ai9 Tomato (Tom) reporter. We have demonstrated that αSMACreER identifies osteoprogenitors in both the periosteum and the bone marrow compartment (*Matthews et al., 2014*; *Grcevic et al., 2012*; *Matthews et al., 2020*). Previously, we delivered tamoxifen at the time of fracture, which potentially labels cells that activate αSMA expression at the time of or shortly after fracture (*Mori et al., 2016*). In this study, we delivered tamoxifen up to 90 days before fracture to evaluate whether the reporter was identifying long-term progenitor cells. First, we confirmed that αSMA-labeled cells were retained in the periosteum over the experimental time course by both histology (*Figure 1A–B*) and flow cytometry (*Figure 1C*). αSMA-labeled cells remained in the periosteum at a similar frequency over the timeframe evaluated and also remained in the endosteum (*Figure 1D*). In juvenile mice (4–5 weeks old), we see contribution of αSMA⁺ cells from the bone marrow compartment, particularly in the trabecular region, to osteoblasts and osteocytes after 17 days (*Figure 1—figure supplement 1A–B*; *Grcevic et al., 2012*). However, 70 days after tamoxifen delivery, osteoblasts are no longer labeled, but some Tom⁺ osteocytes remain (*Figure 1—figure supplement 1C–F*). In contrast, when tamoxifen is

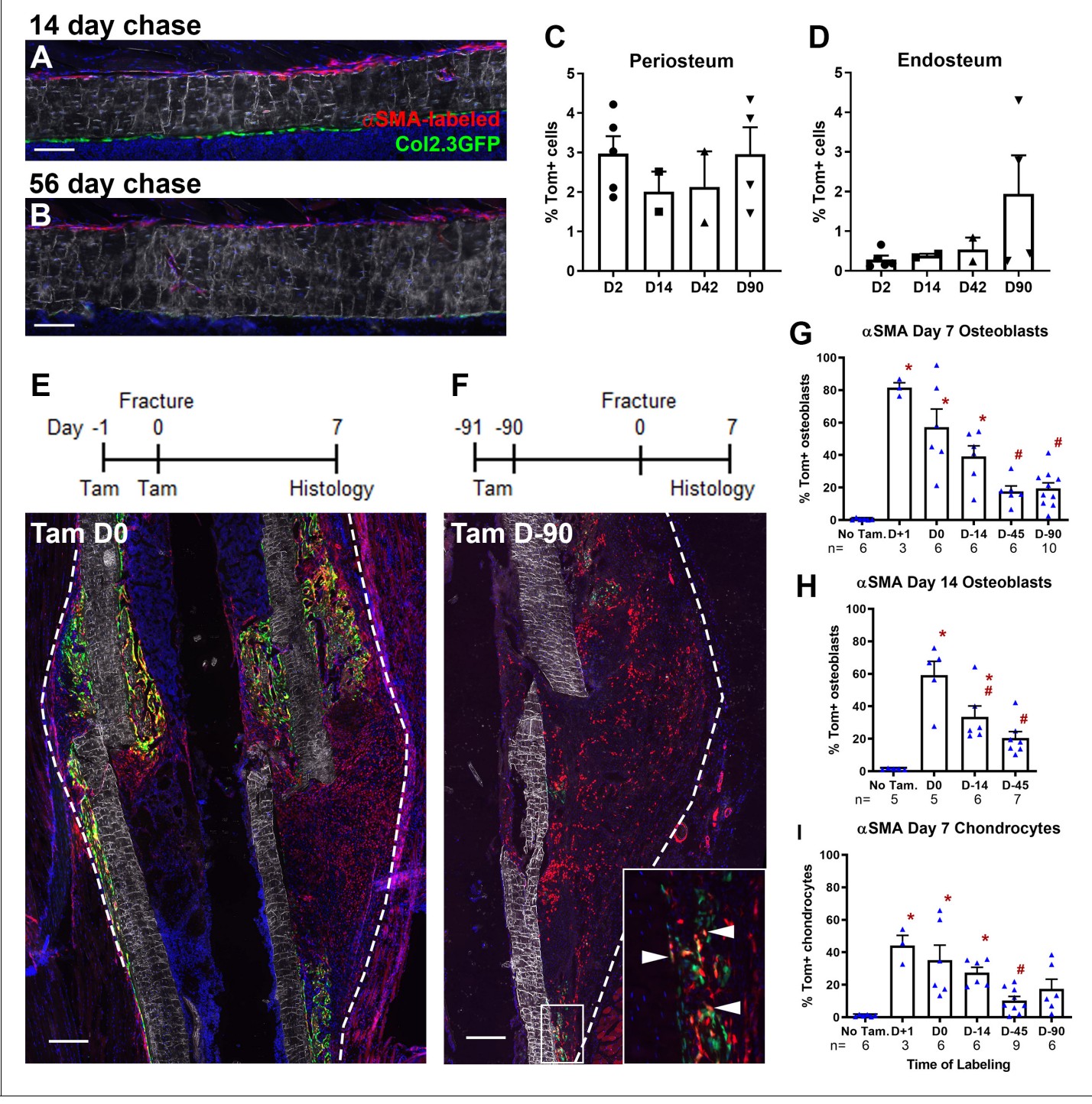

**Figure 1.** Alpha smooth muscle actin (αSMA) identifies long-term progenitor cells in the periosteum. Representative images of tibia mid-shaft periosteum: (A) 14 days or (B) 56 days after tamoxifen with the periosteal surface near the top and endosteal surface near the bottom of the image. Percentage of αSMA-labeled (Tom+) cells within the CD45- population at different time points in (C) periosteum and (D) endosteum as measured by flow cytometry (n = 2–5). Fracture histology of representative day 7 tibial fracture calluses in different groups. Diagrams indicate the experimental design. (E) αSMA-labeled with tamoxifen (Tam) at day −1, 0; (F) αSMA-labeled with tamoxifen at day −91, –90. Quantification of Col2.3GFP+ osteoblasts derived from αSMA-labeled cells in the fracture callus based on image analysis (Tom+GFP+ cells/GFP+ cells) in (G) day 7 fracture callus and (H) day 14 fracture callus at different times after tamoxifen administration. (I) Quantification of Sox9+ chondrocytes derived from αSMA-labeled cells in day 7 fracture callus. All images show mineralized bone by dark field (gray) and DAPI+ nuclei in blue. Scale bars are 100 µm (A, B) or 250 µm (E–F). In F, the boxed area is magnified and αSMA-labeled osteoblasts as quantified in G-H are indicated with arrowheads. Approximate

*Figure 1 continued on next page*

*Figure 1 continued*

callus area used for analysis is indicated by the dashed lines in E-F, cortical bone surface defines the inner boundary. *p<0.05 compared to no tamoxifen control with ANOVA followed by Dunnett's posttest. #p<0.05 compared to tamoxifen day 0 group.

The online version of this article includes the following source data and figure supplement(s) for figure 1:

**Source data 1.** Image analysis processed data and statistical analysis tables (from Graphpad Prism) for graphs in *Figure 1G–I* and supplement 3B-C.

**Figure supplement 1.** Alpha smooth muscle actin (αSMA) lineage tracing in the bone marrow compartment of juvenile mice.

**Figure supplement 2.** Alpha smooth muscle actin (αSMA) lineage tracing in the bone marrow compartment of adult mice.

**Figure supplement 3.** Minimal αSMACreER leakiness in fracture callus and contribution to endosteal bone formation.

delivered to sexually mature mice, there is no contribution to osteoblasts in trabecular bone up to 100 days after tamoxifen (*Figure 1—figure supplement 2*).

Following tibial fracture, αSMA-labeled cells made substantial contributions to fracture callus osteoblasts and chondrocytes (*Figure 1E–I*). Very few Tom$^+$ cells were present or contributed to mature cell types in the absence of tamoxifen (*Figure 1G–I*, *Figure 1—figure supplement 3A*). Most fracture callus osteoblasts are derived from cells that expressed αSMA at some point in their lineage, with 81.5% derived from αSMA$^+$ cells labeled just after fracture. A smaller population of osteoblasts were derived from long-term αSMA-labeled progenitors (around 20%, *Figure 1G–H*). The contribution of αSMA-labeled cells was similar in both the initial intramembranous bone formation in the day 7 callus and the more fully mineralized day 14 callus. Contribution to chondrocytes showed a very similar trend based on the timing of tamoxifen; however, the proportion of Tom$^+$ chondrocytes was always lower (*Figure 1I*). Following the pin insertion and fracture, bone formation also occurs inside the marrow compartment, particularly in the proximal diaphysis surrounding the pin over 1 mm from the fracture site (*Figure 1—figure supplement 3*). αSMA-labeled cells labeled at the time of fracture make a major contribution to newly formed osteoblasts in the marrow compartment; however, there was lower contribution at the later time points suggesting that αSMA identifies long-term injury-responsive osteochondroprogenitors in the periosteum but does not consistently label long-term injury-responsive cells in the bone marrow compartment.

## Putative mesenchymal stem cell markers are enriched in periosteum

We performed further characterization of cell populations in the periosteum, first, to evaluate surface profile using putative mesenchymal stem cell or SSC markers and their corresponding growth potential and, second, to characterize αSMA-labeled cell marker expression. There are numerous markers for SSCs and progenitor cells reported in the literature; however, many have not been evaluated systematically in different tissue compartments. We evaluated the expression of several cell surface markers in periosteum, endosteum, and bone marrow of adult mice (*Figure 2A*, *Figure 2—figure supplement 1*). All stains included a CD45/CD31/Ter119 cocktail to exclude hematopoietic and endothelial cells and select for mesenchymal lineages (termed CD45$^-$ population). We confirmed that the freshly isolated CD45$^+$ population was unable to form fibroblast colony-forming units (CFU-F) or any type of colonies in the CFU-F assay. The CD45$^-$ population in periosteum was highly enriched with most of the markers evaluated compared to both endosteum and bone marrow (*Figure 2A*). Sca1, PDGFRα, and PDGFRβ were abundant in periosteum, but expressed on <2% CD45$^-$ endosteal and bone marrow cells. The only marker that did not follow this trend was CD105, which was consistently present on the majority of endosteal and bone marrow CD45$^-$ cells. Various marker combinations that have previously been used to identify stem cells such as Sca1/PDGFRα or Sca1/CD51 were easily identifiable in periosteum but very rare in the endosteum or bone marrow (*Figure 2—figure supplement 1*). The CD45$^-$ periosteal population formed around 20× more CFU-F than the CD45$^-$ endosteal population, suggesting functional enrichment of stem and progenitor cells (*Figure 2B*). This is in agreement with Duchamp de Lageneste et al. (*Duchamp de Lageneste et al., 2018*) who used a different method to isolate periosteal cells. We went on to evaluate the expansion and differentiation potential of selected periosteal populations independently of αSMA expression using in vitro assays. CFU-F formation was restricted to CD51$^+$ cells, and most enriched in Sca1$^+$ CD51$^+$ cells (*Figure 2—figure supplement 2A*). CD90$^+$ cells also showed high CFU-F potential, whereas CD105$^+$ cells, which are almost exclusively CD90$^-$, formed few or no colonies. Sca1$^+$ CD51$^+$ cells showed the ability to differentiate toward osteogenic (ALP$^+$) and adipogenic (Oil Red O$^+$)

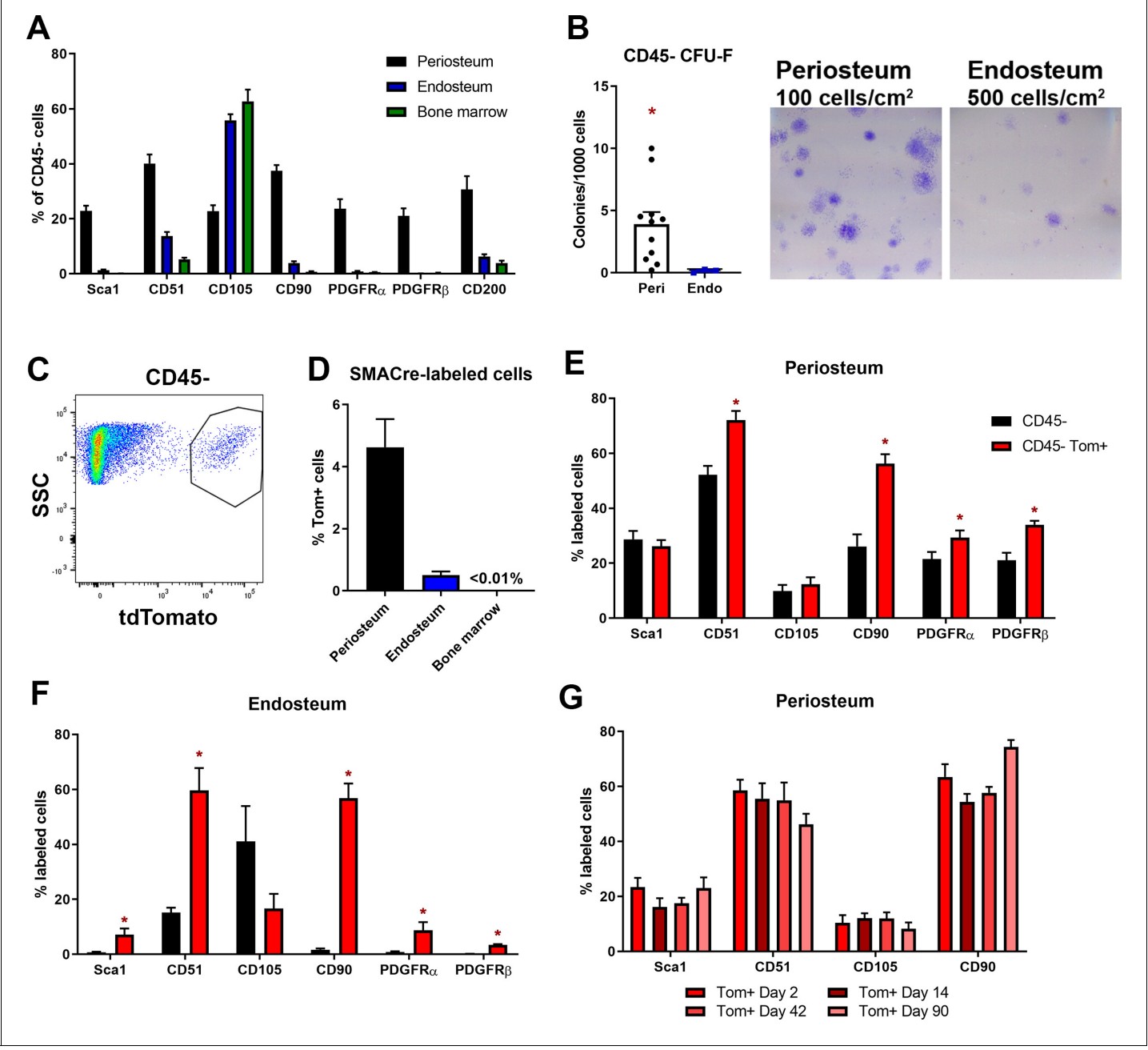

**Figure 2.** Mesenchymal stem cell markers are enriched in periosteum. Flow cytometry and cell sorting was used to evaluate marker expression in tissue from adult long bones. (**A**) Expression of cell surface markers in periosteum, endosteum, and bone marrow samples as a percentage of the CD45/Ter119/CD31$^-$ fraction, n = 4–50. Expression in periosteum is significantly different from both bone marrow and endosteum for all markers (two-way ANOVA with posttest comparing cell types). (**B**) Colony-forming unit fibroblast (CFU-F) formation in the total CD45/Ter119/CD31$^-$ fraction. (**C**) Representative dot plot showing gating of alpha smooth muscle actin (αSMA)-labeled cells. (**D**) Proportion of αSMA-labeled cells within the CD45/Ter119/CD31$^-$ gate 2–3 days after tamoxifen in different tissue compartments (n = 17). Cell surface marker expression in the Tom$^+$ population 2–3 days after tamoxifen in (**E**) periosteum and (**F**) endosteum (n = 4–17). (**G**) Cell surface profile of αSMA-labeled periosteum cells at different time points (n = 2–7). *p<0.05 compared to CD45$^-$ population.

The online version of this article includes the following source data and figure supplement(s) for figure 2:

**Source data 1.** Data tables and statistical analysis tables (from Graphpad Prism) for graphs shown in *Figure 2* and figure supplement 2.

**Figure supplement 1.** Flow cytometry plots of mesenchymal progenitor marker expression.

**Figure supplement 2.** Sca1$^+$CD51$^+$ and CD90$^+$ periosteal mesenchymal cells are enriched for multipotent progenitor cells.

lineages, in many cases within a single colony (*Figure 2—figure supplement 2B-C*). Similar results were obtained for the CD90$^+$ population. Sca1$^-$ CD51$^+$ cells and CD90$^-$ CD105$^-$ cells efficiently activated ALP but never differentiated into adipocytes. This suggests that in addition to including some mature osteoblasts, these populations contain committed osteoprogenitors that remain capable of density-independent growth. All these populations contained a small proportion of αSMA-labeled cells, with the highest frequency of αSMA-labeled cells in the CD90$^+$ population (*Figure 2—figure supplement 2D*).

We next performed further characterization of αSMA-labeled cells in the absence of injury. By flow cytometry, we confirmed that αSMA$^+$ cells were much more prevalent in periosteum compared to endosteum and bone marrow in adult mice (*Figure 2C–D*), although they were rarer than the other markers we analyzed. We evaluated expression of various markers in the αSMA$^+$ population. The majority of αSMA-labeled cells in the periosteum and endosteum express CD51 and CD90 (*Figure 2E–F*). The other markers tested were expressed in smaller subsets of αSMA-labeled cells, indicating there is heterogeneity within the labeled population. We next evaluated whether the prevalence of any of these markers in the αSMA-labeled population changed over time since labeling. We did not detect differences in the level of expression of any of the markers evaluated individually or in various combinations at different times after tamoxifen, suggesting that additional markers may be required to define the heterogeneity and phenotype of these cells (*Figure 2G*).

To further evaluate heterogeneity, we performed plate-based single cell RNAseq analysis on αSMA-labeled periosteal cells isolated at different time points after tamoxifen administration (*Figure 3*). We obtained data from 185 cells from 2 days after tamoxifen, 197 cells from 6 weeks, and 147 cells from 13 weeks after tamoxifen. The median unique molecular identifier (UMI) number was 697, and the median number of expressed genes was 449 (*Figure 3—figure supplement 1*). Since expression levels for the majority of genes were low in this study, we restricted our analysis to basic clustering studies. Clustering using the Leiden community detection algorithm generated four clusters from the αSMA-labeled cells, confirming heterogeneity in this population (*Figure 3A*, *Figure 3— figure supplement 2*; *Traag et al., 2019*). Clusters 1–3 are similar and appear to represent mainly periosteal populations. Cluster 4 is quite separate from the rest and appears to be derived from muscle contamination, expressing markers such as skeletal muscle actin, *Acta1*, and myoglobin, *Mb* (*Figure 3—figure supplement 2B*). We have previously demonstrated that αSMA labels a subset of muscle satellite cells that go on to generate myofibers in both juvenile and adult animals (*Matthews et al., 2016*). Notably, of the 26 cells in this cluster, only one was from the day 2 chase group, consistent with the differentiation of αSMA$^+$ satellite cells into mature muscle lineage cells over time.

Cluster 3 is characterized by expression of a number of pericyte markers, including *Rgs5*, and frequent expression of PDGFRβ and αSMA. They also frequently express *Cxcl12*, which is a marker of bone marrow resident reticular cells, and *Fabp4*, which is known as an adipocyte lineage marker (*Figure 3B*, *Figure 3—figure supplement 2A*). Other adipocyte markers like *Adipoq* and *Pparg* were undetectable in most cells. The proportion of cluster 3 cells increased slightly over time, suggesting they might represent a population of mature perivascular cells (*Figure 3C*). Clusters 1 and 2 show some overlap, and cluster 1 had few selective markers, with only the matrix protein gelsolin, *Gsn*, passing our marker criteria. Interestingly, cluster 1 shows the most frequent expression of the genes for Sca1, PDGFRα, and CD90, suggesting it may contain multipotent stem/progenitor populations (*Figure 3D*). The proportion of cells in cluster 1 was reduced following longer chase periods, in line with the reduction in αSMA-lineage contribution to fracture healing at later time points. Markers of cluster 2 are primarily matrix genes that can be expressed by the osteoblast lineage, including *Col1a2*, *Tnmd*, *Fmod,* and *Ogn*. Unpublished studies we performed in parallel with differentiated osteoblasts show consistently high levels of *Bglap* in differentiated osteoblasts, but *Bglap* is absent or present at only low levels in the αSMA-labeled cells, indicating that cluster 2 does not represent mature osteoblast lineage cells. It is feasible that at least some of these cells are from the fibrotendonous lineage, as the periosteum contains fascia and fibrous tendon attachments, which appear to be targeted by αSMA histologically. Overall, our data indicate that αSMA$^+$ cells are a fairly rare, but heterogeneous cell population within the periosteum that can be separated into at least three different cell types.

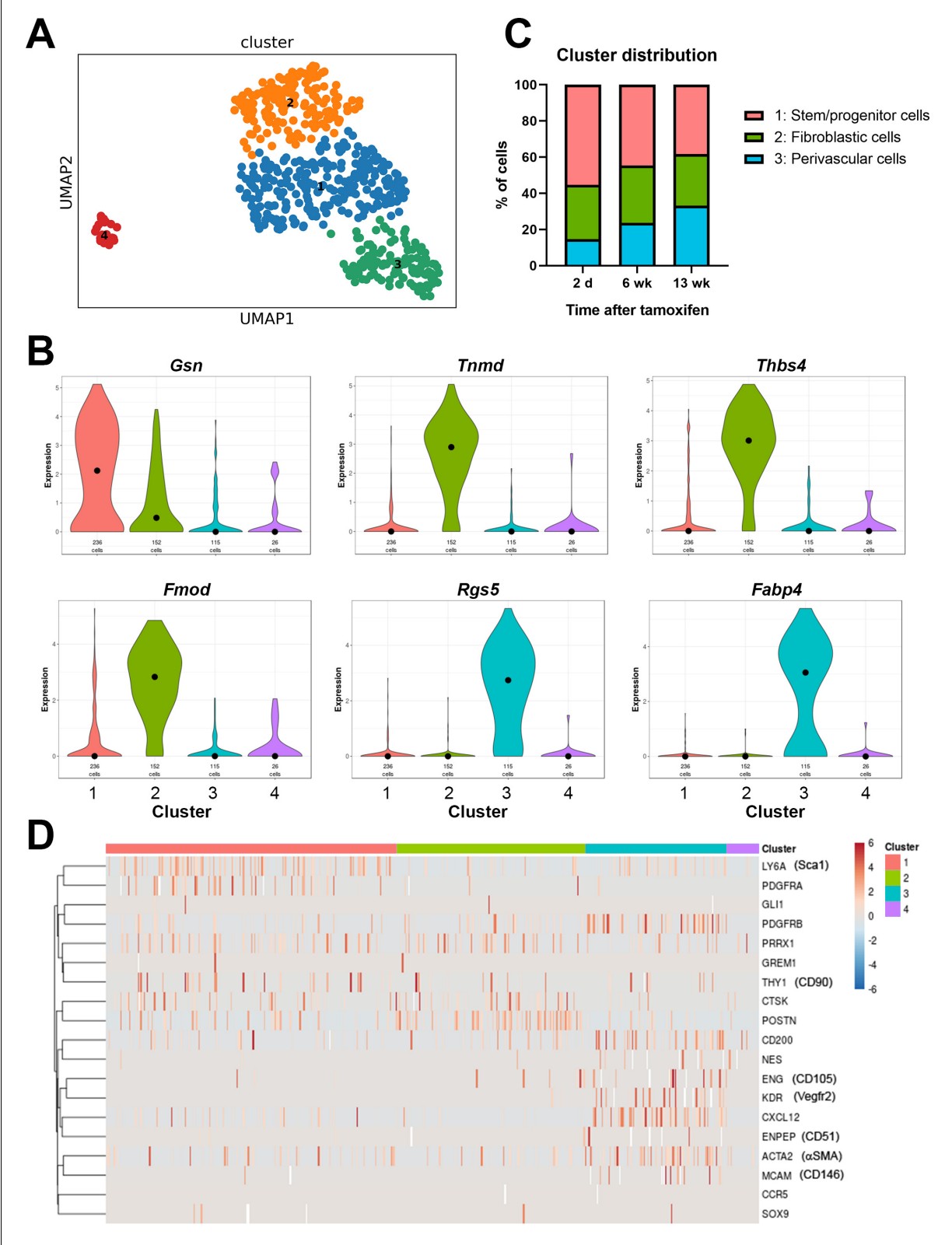

**Figure 3.** Single cell RNAseq analysis of αSMA-Tom⁺ cells. Plate-based RNAseq was performed on alpha smooth muscle actin (αSMA)-Tom⁺ cells at up to 13 weeks after tamoxifen delivery. (A) UMAP cluster plot indicating four clusters. (B) Violin plots of top markers for clusters 1–3. (C) Distribution of cells in clusters 1–3 at different time points following tamoxifen. (D) Heat map of expression of putative skeletal stem and periosteal progenitor markers.

*Figure 3 continued on next page*

*Figure 3 continued*

The online version of this article includes the following source data and figure supplement(s) for figure 3:

**Source data 1.** scRNAseq cluster makers with AUROC values >0.7.
**Figure supplement 1.** Quality control data for single-cell RNAseq dataset.
**Figure supplement 2.** Heat maps of cluster markers.

## Label-retaining cells in the periosteum are enriched with progenitor markers

One feature of long-term stem cells is a slow cycling rate, and this can be evaluated experimentally using label retention assays. We used a doxycycline (dox)-inducible GFP-tagged histone system, H2B-GFP, which is incorporated into the DNA during dox feeding, then diluted during subsequent cell division. This system has been used for enrichment of highly self-renewing populations of hematopoietic stem cells and muscle satellite cells (*Foudi et al., 2009*; *Chakkalakal et al., 2014*). The experimental design for generating label-retaining cells (LRCs) is shown in *Figure 4A*. H2B-GFP was incorporated into the majority of periosteal cells (*Figure 4B*, *Figure 4—figure supplement 1A*). Quiescent terminally differentiated cells such as osteocytes also effectively retain the label (*Figure 4—figure supplement 1B*). Flow cytometry was used to track GFP expression in the CD45⁻ fraction of the periosteum (*Figure 4B–C*). Our periosteal isolation procedure excludes bone tissue and therefore osteocytes, and we have previously demonstrated that our endosteal preparations contain almost no osteocytes (*Matic et al., 2016*). After the chase phase, the periosteum contained a small population of true LRCs (GFP$^{hi}$), as well as a slightly larger GFP$^{int}$ population which would be expected to contain some LRCs. Histologically, the LRCs were mostly in the inner cambium layer of the periosteum (*Figure 4—figure supplement 1B*). We confirmed that most cells capable of forming CFU-F initially incorporated a GFP label (*Figure 4D*). After the chase, both GFP$^+$ populations showed enrichment of CFU-F compared to the GFP$^-$ fraction (*Figure 4E*). We also evaluated cell surface marker expression in LRCs and found strong enrichment of Sca1 and CD51, more frequent expression of CD90, and fewer CD105$^+$ cells in periosteum (*Figure 4F*) and endosteum (*Figure 4G*) compared to GFP$^-$ cells. Since label retention alone was not very effective at enriching CFU-F, we combined this approach with αSMA-labeling. Initially, we labeled 73.4 ± 7.6% of αSMA$^+$ cells, and after at least 13 weeks chase, 22.6 ± 7.7% were LRCs, indicating a large proportion of these cells are quiescent (*Figure 4—figure supplement 2A*). This is a substantially higher proportion of LRCs than we found in the overall CD45⁻ population of the same samples (3.2 ± 1.5%, p<0.0001). Ex vivo, αSMA-labeled LRCs (GFP$^+$Tom$^+$) were enriched for CFU-F compared to the GFP-Tom$^+$ population (*Figure 4—figure supplement 2B*). Interestingly, αSMA-labeled LRCs appeared to be lineage-restricted, with evidence of differentiation along the osteogenic lineage but absence of adipogenic differentiation (*Figure 4—figure supplement 2C*). αSMA-labeled LRCs were more frequently CD51$^+$, but fewer were CD90$^+$ that Tom$^+$ non-LRCs (*Figure 4—figure supplement 2D*).

## Evaluation of SSC populations in adult mice

In order to systematically evaluate stem cell populations in adult animals, we evaluated the utility of the stain described by Chan et al. to identify SSC and BCSP (*Chan et al., 2015*). Stem/progenitor markers and gating strategies should, by definition, exclude mature cell types of the lineage. There are no validated methodologies available to identify osteoblasts using cell surface markers, although some authors report Sca1$^+$CD51$^+$ as stem cells and Sca1-CD51$^+$ cells as committed osteoprogenitors or osteoblasts (*Arthur et al., 2019*; *Lundberg et al., 2007*). We utilized the Col2.3GFP transgene to identify mature osteoblasts (*Kalajzic et al., 2002*; *Dacquin et al., 2002*). Col2.3GFP$^+$ cells were always present in endosteal samples (5.4 ± 5.0% of CD45-) and were detectable in numbers sufficient for analysis in some periosteum samples (1.3 ± 1.3% of CD45-). CD200 expression was present on over 90% of Col2.3GFP osteoblasts in both endosteum and periosteum, suggesting that CD200 may be useful for positive selection of osteoblasts. Around 35% of endosteal osteoblasts were CD51+ (*Figure 5A*; *Matic et al., 2016*). A subset of osteoblasts in both tissues were CD105$^+$, while the other markers including Sca1, CD90, and Ly51 (also known as 6C3) were rare or absent in endosteal osteoblasts. The gating strategy to identify SSCs and BCSPs in both periosteum and endosteum is shown in *Figure 5C–D*. We found that Col2.3GFP$^+$ osteoblasts constituted almost 40% of

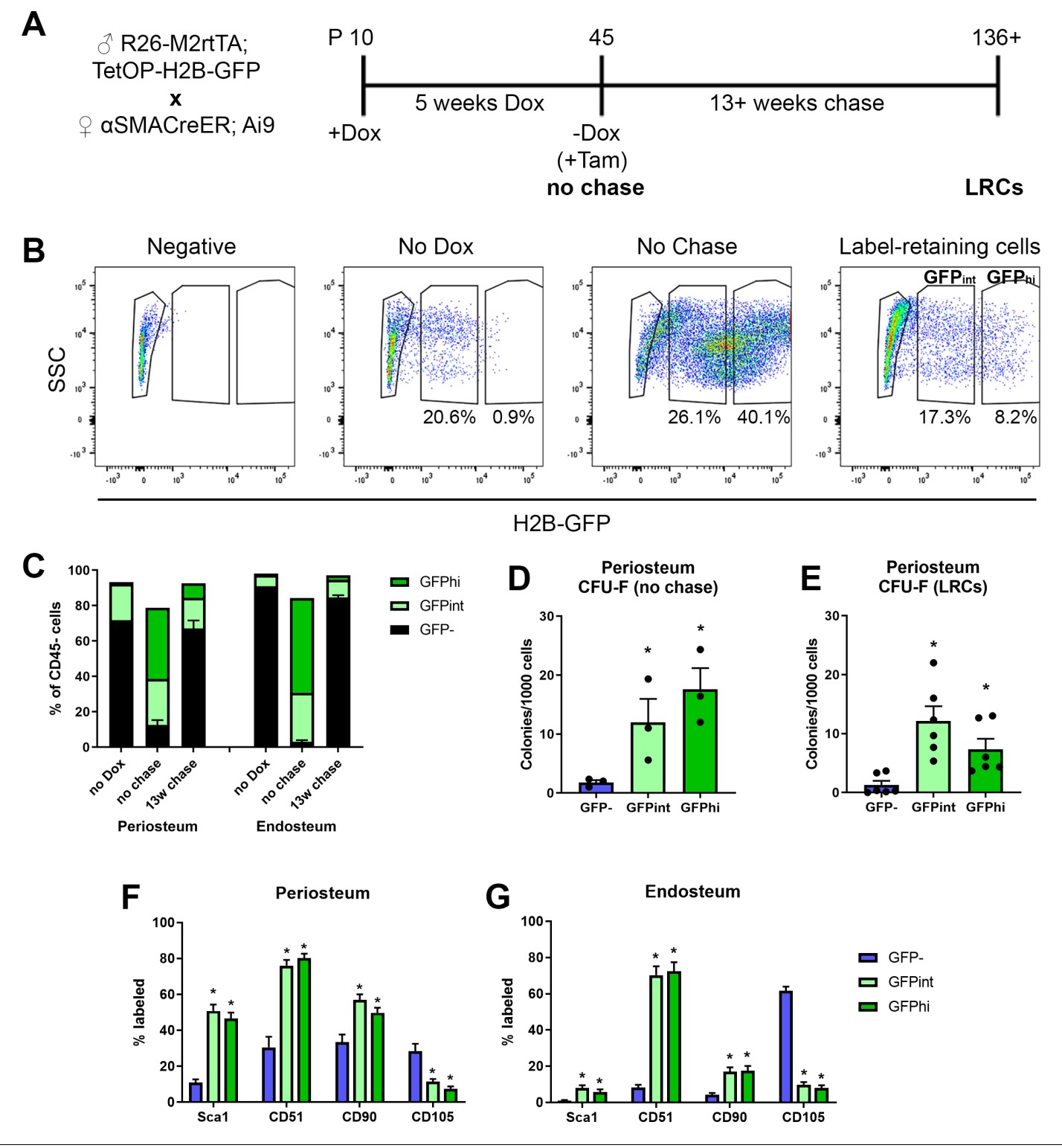

**Figure 4.** Label-retaining cells are enriched with most progenitor markers. (**A**) Experimental design for label-retaining cell experiments. (**B**) Representative gating of the CD45⁻ population in different periosteum samples to define three populations based on GFP expression. (**C**) Quantification of different GFP populations in a representative experiment for periosteum and endosteum (n = 1, no doxycycline [dox]; n = 4, no chase; n = 6, 13-week chase). (**D**) Colony-forming unit fibroblast (CFU-F) frequency in different CD45⁻ GFP populations in periosteal no chase samples (n = 3) and (**E**) following chase (n = 6). Cell surface marker expression is shown in different CD45⁻ GFP subsets following chase (populations equivalent to right-
*Figure 4 continued on next page*

*Figure 4 continued*

hand label-retaining cell plot in B) in (F) periosteum cells and (G) endosteal cells. Data is pooled from three separate experiments involving 13–22 weeks chase (n = 19). *p<0.05, repeated measures ANOVA with Dunnett's posttest compared to GFP⁻ population.

The online version of this article includes the following source data and figure supplement(s) for figure 4:

Source data 1. Data tables and statistical analysis tables (from Graphpad Prism) for graphs shown in *Figure 4F–G*.
Figure supplement 1. Histology of H2B-GFP/αSMACreER/Ai9 femurs.
Figure supplement 2. Alpha smooth muscle actin (αSMA)-labeled periosteal label-retaining cells are enriched with colony-forming unit fibroblast (CFU-F).

endosteal cells defined as SSC (CD45-CD51$^+$CD90-Ly51-CD105-CD200$^+$) and 20% defined as BCSP (CD45-CD51$^+$CD90-Ly51-CD105$^+$) (*Figure 5D*). The proportion of mature osteoblasts in these populations was lower in periosteal samples but still represented 7–8% of SSC and BCSP (*Figure 5C*). The pre-BCSP population (CD45-CD51$^+$CD90-Ly51-CD105-CD200-) contains very few osteoblasts. In addition, when we evaluate the proportion of SSC and BCSP in different tissue compartments, we find that the GFP$^+$ osteoblast populations, particularly in the endosteum, have the highest frequency of these 'stem cells' (*Figure 5B*). Therefore, these marker combinations used for SSC and BCSP do not exclude mature cells within the osteoblast lineage in adult mice, so do not specifically identify stem cells in this setting.

## αSMA identifies transplantable, self-renewing osteoprogenitors

In ex vivo assays, αSMA$^+$ periosteal cells were capable of CFU-F formation (*Figure 6A*). Following transplantation into a critical-sized calvarial defect, αSMA$^+$ cells were readily identified in all sections evaluated, indicating the ability to engraft and expand in the defect (*Figure 6B*). Some cells activated Col2.3GFP, indicating limited differentiation into osteoblasts, similar to a recent study (*Ortinau et al., 2019*). We did not see evidence of cartilage formation in these defects at the time point evaluated. We tested the in vivo self-renewal potential of αSMA-labeled cells by performing a fracture, allowing it to heal for 8 weeks, then refracturing the same bone in a similar location. When tamoxifen was given at the time of initial fracture, around 20% of both osteoblasts and chondrocytes in the secondary fracture callus are derived from the original αSMA-labeled population (*Figure 6C–J*). This is 35–50% of the contribution in the first fracture but indicates self-renewal of a proportion of αSMA-labeled progenitors. Contribution to the secondary fracture when tamoxifen was given 14 days prior to the first fracture was not statistically significantly different to the day 0 group, confirming identification of self-renewing osteochondroprogenitors.

## Functional contribution of αSMA$^+$ cells to fracture healing

We evaluated the functional importance of αSMA+ cells during fracture healing using the Rosa-DTA model of cellular ablation. Diphtheria toxin (DTA) expression, which causes death of the cells where it is expressed, was driven by αSMACreER and induced by tamoxifen delivery around the time of fracture. When ablation was performed at the time of fracture (days 0, 1), we saw a 50% reduction in αSMA-labeled cells in 4 days post-fracture (DPF) callus tissue by flow cytometry (*Figure 7A*). This is consistent with the 50% efficiency of recombination we demonstrated for αSMACreER in vitro (*Sinder et al., 2020*). Histologically, there is a clear reduction in fracture size and αSMA-labeled cells at 4 DPF (*Figure 7B*). Histological analysis at 14 DPF revealed a significant reduction in total callus size as well as mineralized area (*Figure 7C–D*). Micro-computed tomography (microCT) results at day 14 confirmed the histological observations (*Figure 7E*). At 21 DPF, microCT showed no change in total callus volume but a significant decrease in callus bone mass (*Figure 7F*). Similar impairments in healing were seen whether ablation was performed at the time of fracture or throughout the first 8 days of healing, suggesting that early loss of αSMA+ cells was most important for this phenotype. Overall, our data demonstrate that partial ablation of αSMA+ progenitors reduced callus formation and delayed mineralization. The less severe phenotype at 21 DPF suggests this may be a delay in healing rather than complete disruption of the process, although the fractures from mice with αSMA-ablation probably never reach the size of control calluses. Healing in the DTA mice may still progress due to compensation of αSMA+ progenitors that escaped ablation or larger contribution from αSMA-negative progenitors in this setting.

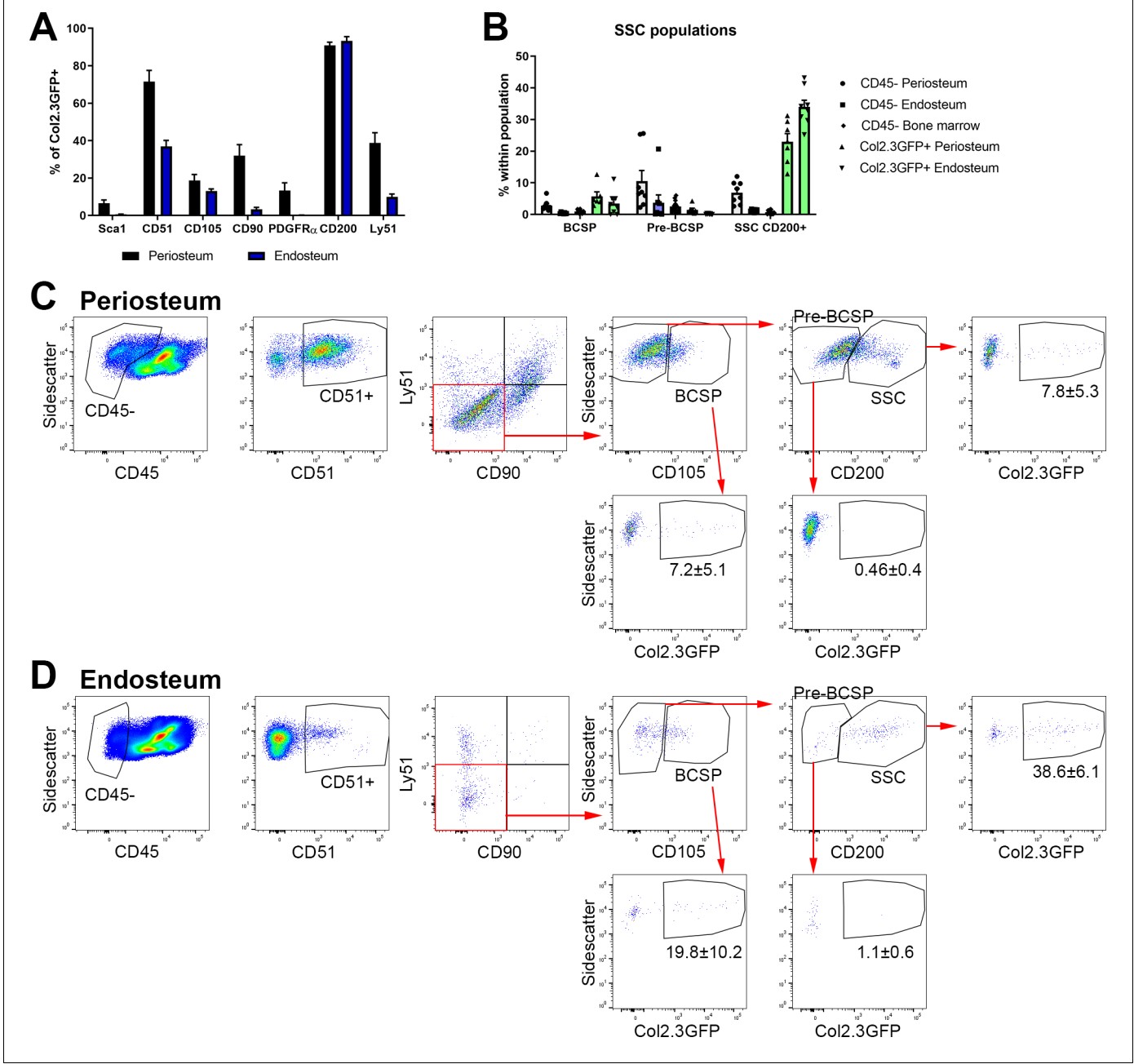

**Figure 5.** Bone, cartilage, and stromal progenitor (BCSP) and skeletal stem cell (SSC) populations in adult bone contain mature osteoblasts. (A) Marker expression on Col2.3GFP[+] osteoblasts, n = 4–32. (B) Frequency of BCSP and SSC populations in different tissue compartments and populations (n = 8–12). (C–D) Gating to identify BCSPs, pre-BCSPs, SSCs, and presence of Col2.3GFP[+] osteoblasts as a major component of these populations in periosteum (C) and endosteum (D). Mean ± standard deviation for % GFP[+] cells are shown.

The online version of this article includes the following source data for figure 5:

**Source data 1.** Data tables for graphs and plots shown in *Figure 5*.

## Contribution of other periosteal populations to fracture healing

We have demonstrated that αSMA identifies some, but not all, of the cells contributing to osteoblasts and chondrocytes in the fracture callus. We therefore examined the potential contribution of other populations. We tested the previously unexplored hypothesis that mature osteoblast lineage cells may contribute to osteoblast formation in the fracture callus. Col2.3CreER-labeled cells made a small but significant contribution to osteoblast formation in the fracture callus, but only when

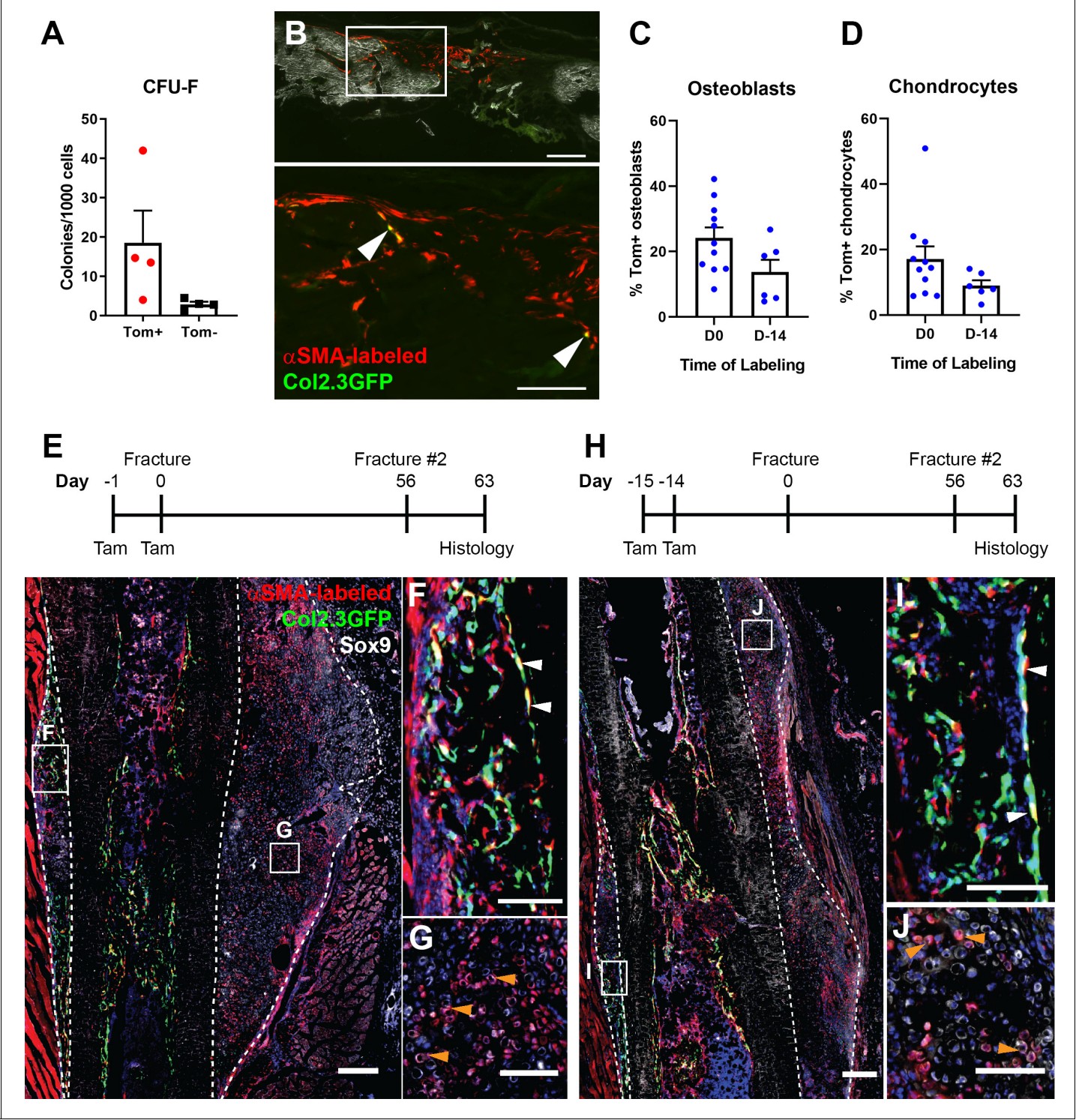

**Figure 6.** Alpha smooth muscle actin (αSMA)-labeled cells are transplantable self-renewing osteochondroprogenitors. (A) Colony-forming unit fibroblast (CFU-F) frequency of periosteal αSMA-labeled cells sorted 2 days after tamoxifen. (B) Engraftment of transplanted αSMA-labeled cells in a calvarial defect 5 weeks after transplantation. Arrowheads indicate cells that are Col2.3GFP+. Quantification of αSMA-labeled cell contribution to (C) osteoblasts and (D) chondrocytes in day 7 secondary fracture calluses is shown. (E) Experimental design and fracture callus histology following a second tibia fracture in αSMACreER/Ai9/Col2.3GFP mice showing contribution of Tom+ cells to callus tissue, including osteoblasts (F) and Sox9+ chondrocytes (white) (G). (H–J) Histology of a similar secondary fracture where tamoxifen was given 14 days prior to the primary fracture. White arrowheads indicate

*Figure 6 continued on next page*

*Figure 6 continued*

αSMA-labeled Col2.3GFP+ osteoblasts, orange arrowheads indicate αSMA-labeled Sox9+ chondrocytes. DAPI stain is shown in blue. Approximate callus area used for analysis is indicated by the dashed lines in E and H. Scale bars indicate 250 μm (low magnification) and 100 μm (high magnification). The online version of this article includes the following source data for figure 6:

**Source data 1.** Data tables and statistical analysis tables (from Graphpad Prism) for graphs shown in *Figure 6C-D*.

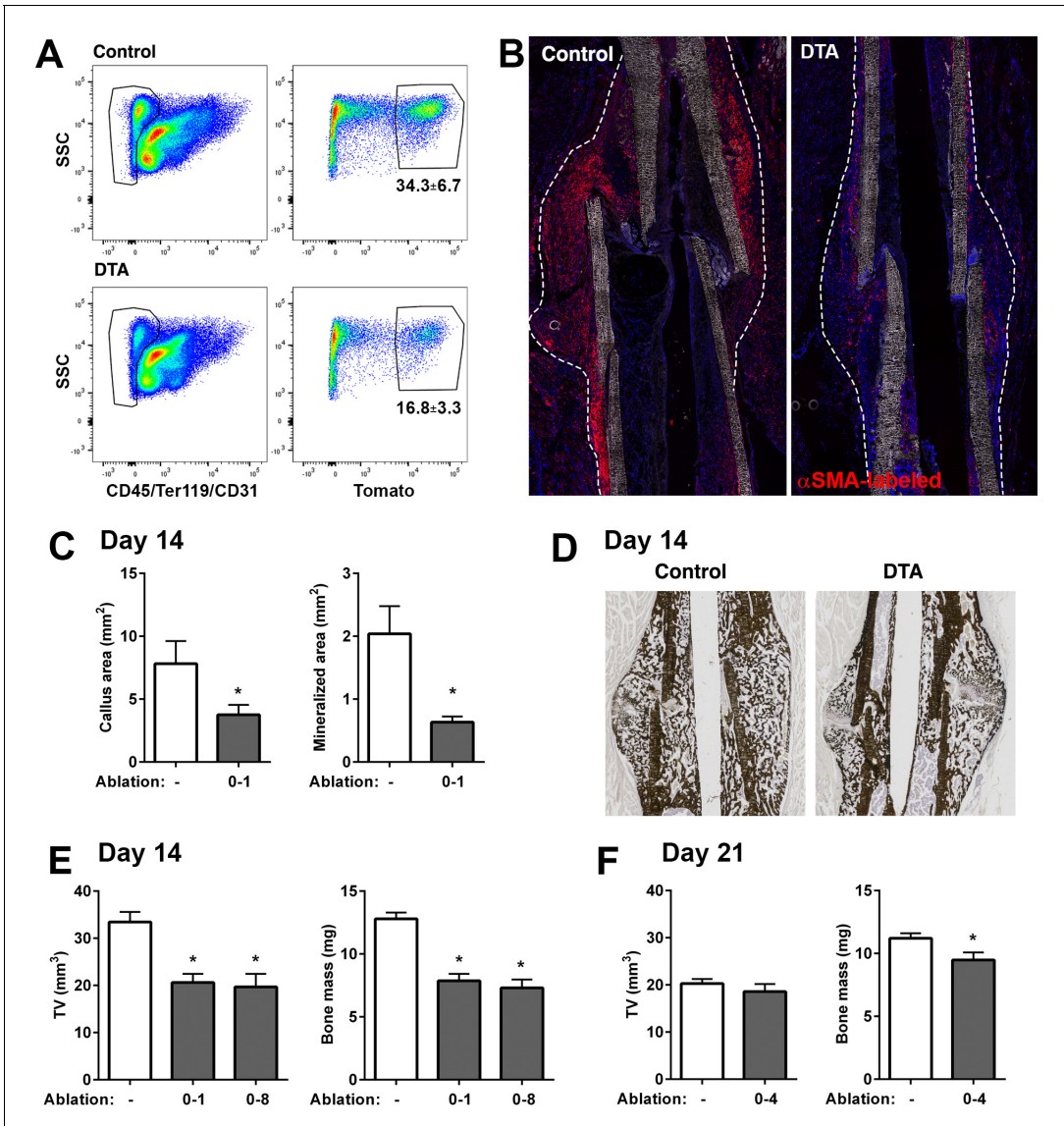

**Figure 7.** Ablation of alpha smooth muscle actin (αSMA)+ cells impairs fracture healing. (A) Flow cytometry of day 4 fracture callus from control αSMACreER/Ai9 mice and DTA animals (αSMACreER/Rosa-DTA/Ai9) indicating significantly fewer (p<0.01) αSMA-labeled cells in the ablation model (mean ± SD shown, n = 4). (B) Representative histological sections from day 4 fracture callus in controls or with DTA-mediated ablation. The outline of the callus is indicated. (C) Histological analysis of callus area and mineralized area at day 14, and (D) representative images of von-Kossa staining (n = 4–7). (E) Micro-computed tomography (MicroCT) analysis of day 14 fracture callus in DTA⁻ and DTA⁺ mice that had undergone two different tamoxifen regimens to ablate αSMA+ cells, day 0, 1 of fracture or day 0, 2, 4, 6, 8. Callus tissue volume and bone mass were measured. Cre-: n = 14, 8 male (M), 6 female (F); DTA D0-1: n = 10, 4M 6F; DTA D0-8: n = 8, 4M 4F. (F) MicroCT analysis of day 21 fracture callus in mice treated with tamoxifen on day 0, 2, and 4 (n = 14–15).

The online version of this article includes the following source data for figure 7:

**Source data 1.** Data tables and statistical analysis tables (from Graphpad Prism) for graphs shown in *Figure 7C,E and F*.

tamoxifen was given close to the time of fracture (*Figure 8.A*-B). This is in line with previous studies that estimate a lifespan of around 2 weeks for osteoblasts (*Matic et al., 2016*; *Jilka et al., 1998*). We did not detect any contribution of Col2.3-labeled cells to chondrocytes. This suggests that committed osteoblast lineage cells present in the periosteum can contribute to fracture osteoblasts but do not have the plasticity to contribute to chondrocytes.

Finally, we further evaluated PDGFRα as an alternate marker of periosteal progenitor cells as we hypothesized that it may target a broader periosteal progenitor population than αSMA. PDGFRaCreER targeted the majority of periosteal cells 1 day after completion of tamoxifen administration, as well as the majority of osteoblasts on trabecular and endocortical bone surfaces (*Figure 8C–E*). It also targeted a population of cells in the bone marrow. This labeling was tamoxifen-dependent (*Figure 8F*). These results were confirmed with immunostaining for PDGFRα that indicated expression in the periosteum and in endosteal osteoblasts (*Figure 8G*). There is some discrepancy between our flow data and this result (see *Figure 2A* and *Figure 5A*). PDGFRα staining by flow cytometry tends to be weak, and there is rarely a clear demarcation between positive and negative cells (*Figure 2—figure supplement 1*). In contrast, PDGFRα-driven Cre in combination with a strong reporter gene such as Ai9 amplifies even low transgene activity. It is therefore likely that we are only gating for cells highly expressing PDGFRα in flow cytometry, but this makes it difficult to use as a reliable marker. This discrepancy is consistent with the results of Ambrosi et al. who report much wider expression of PDGFRα using a GFP reporter mouse compared to previous results with antibody staining and flow cytometry (*Morikawa et al., 2009*; *Ambrosi et al., 2017*). Their data also suggests expression of this reporter in osteoblasts. Another recent study reported much higher PDGFRα expression in bone marrow compared to periosteum in contrast with our flow cytometry results (*Tournaire et al., 2020*). We conclude that PDGFRα is not an optimal marker of progenitor cells within the periosteum due to both technical variability depending on the method used and expression in mature osteoblasts.

## Discussion

Our results indicate that the periosteum of adult mice is highly enriched for cells with markers and phenotype of stem and progenitor cells. Together with other recent studies, our results highlight the presence of unique cell populations within the periosteum that are rare or absent in the bone marrow compartment, and demonstrate that stem cell markers can change with age (*Debnath et al., 2018*; *Duchamp de Lageneste et al., 2018*; *Ortinau et al., 2019*; *Tournaire et al., 2020*). There is abundant evidence suggesting that fracture healing is impaired if periosteum is removed, or that otherwise critical-sized defects in a variety of bones can regrow if periosteum is retained (*Mashimo et al., 2013*; *Kuwahara et al., 2019*; *Ozaki et al., 2000*; *Zhang et al., 2005*). There are also numerous studies reporting superior outcomes with cultured periosteal cells compared to other cell sources like bone marrow in terms of bone formation both in vitro and in vivo (*Duchamp de Lageneste et al., 2018*; *van Gastel et al., 2014*). Of the markers we examined, we found that most were much more abundant in the periosteum than endosteum. Sca1, CD51, CD90, αSMA, and PDGFRα (*Wang et al., 2019a*) all enrich for CFU-F in periosteum, similar to what has been reported in other cell types. Although we did not thoroughly characterize PαS cells in the periosteum, we note that these cells were very rare in the endosteal compartment. Morikawa et al. report large enrichment of PαS cells following collagenase digestion of crushed bones (*Morikawa et al., 2009*). Our results raise the possibility that some of the cells isolated in their studies were actually derived from the periosteum. We confirmed by immunostaining and use of an inducible Cre that PDGFRα expression was very abundant in the periosteum. This result and abundant PDGFRα expression shown histologically in osteoblasts were not completely replicated in our flow cytometry data. Ambrosi et al. reported that all Sca1+ cells in the bone marrow co-express PDGFRα using a PDGFRα-GFP line which contrasted with the original study characterizing PαS cells (*Ambrosi et al., 2017*). This highlights the difficulty in using PDGFRα consistently as a stem and progenitor cell marker. With respect to CD51, our results agree with the studies using neonatal bones, indicating that progenitor cells are solely contained within the CD51+ population (*Chan et al., 2009*); however, a significant proportion of mature osteoblasts are CD51+, so additional markers that exclude differentiated cells are required. Finally, our results suggest that strong Sca1 expression is specific to periosteal mesenchymal cells and enriches multipotent stem or progenitor cells in combination with

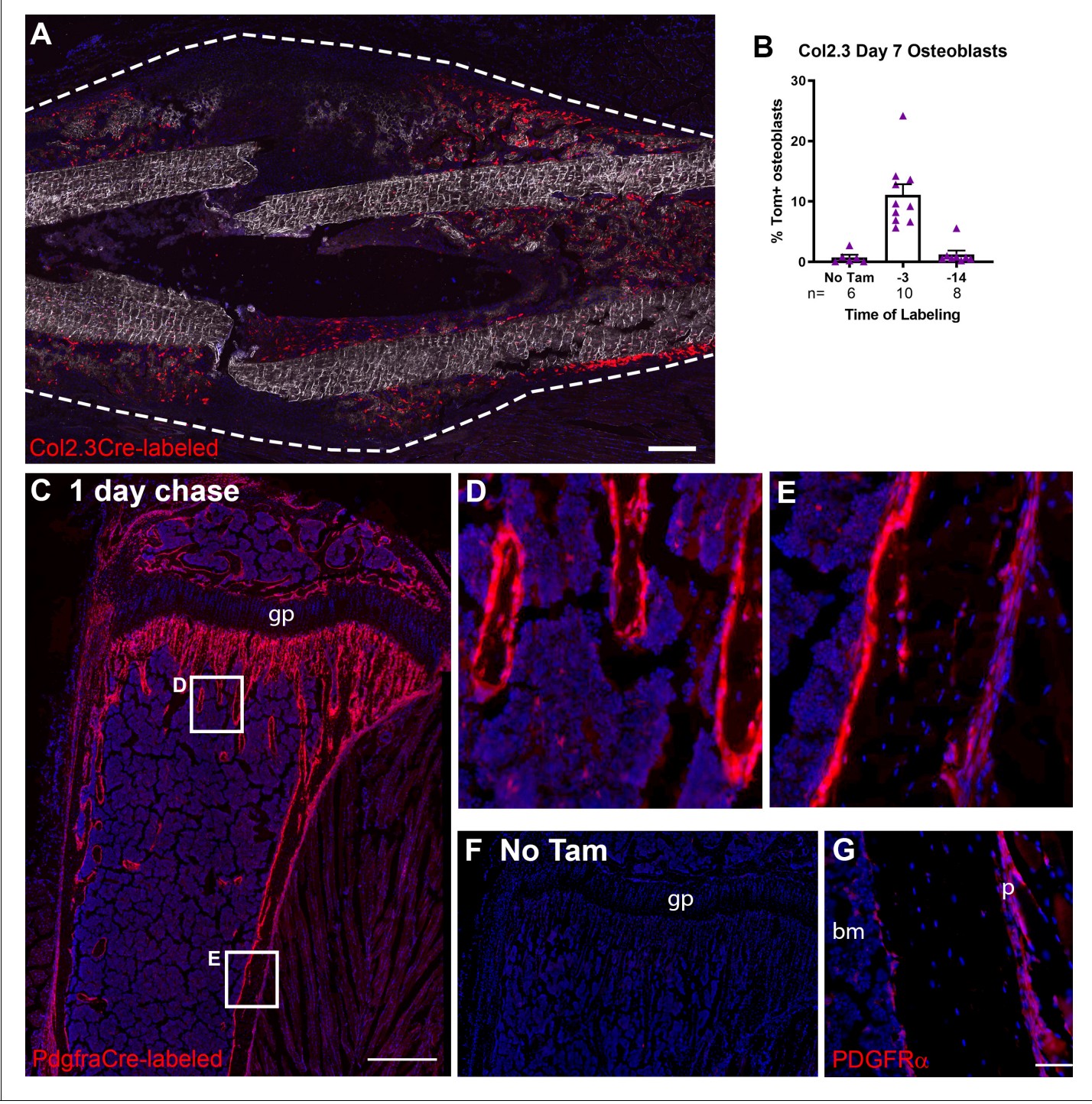

**Figure 8.** Contribution of mature osteoblasts to fracture healing and PDGFRα distribution. (**A**) Fracture histology of a representative day 7 tibial fracture callus from a Col2.3CreER/Ai9 mouse treated with tamoxifen at days −4 and −3. Mineralized bone is shown by dark field (gray) and DAPI+ nuclei in blue. Scale bar is 250 µm. (**B**) Quantification of Col2.3-labeled cell contribution to Osx+ osteoblasts. *p<0.05 based on ANOVA with Dunnett's posttest compared to no tamoxifen group. (**C**) Tibia histology of PdgfraCreER/Ai9 mice given three daily doses of tamoxifen and euthanized 1 day later. Scale bar is 200 µm. Magnified areas of (**D**) trabecular and (**E**) cortical bone are shown. (**F**) Tibial histology of a PdgfraCreER/Ai9 mouse that had not been exposed to tamoxifen showing minimal leakiness of the CreER. Representative images of n = 3 aged 5–8 weeks of age are shown. (**G**) PDGFRα immunostaining of bone. Scale bar is 50 µm. bm, bone marrow; gp, growth plate; p, periosteum.

The online version of this article includes the following source data for figure 8:

**Source data 1.** Data table and statistical analysis table (from Graphpad Prism) for graph shown in *Figure 8B*.

CD51, at least in vitro. Sca1 expression was also highly enriched in LRCs in all bone tissue compartments. A recent study using scRNAseq in periosteal cells suggested Sca1 as a marker of differentiated cells in the periosteum, although it was unclear what cell type these mature cells were (*Debnath et al., 2018*). Inducible Cre models driven by expression of the Sca1 gene are now available and would potentially help to confirm the in vivo function of Sca1+ cells. Notably, they target endothelial progenitors which will complicate any analysis with these models (*Vagnozzi et al., 2018*; *Tang et al., 2018*), and one of these models was reported to target very few cells within the bone (*Vesprey et al., 2020*).

LRCs in the periosteum showed large differences in cell surface marker profile compared to non-LRCs and enrichment of CFU-F formation. However, the experimental system presented some limitations, as we did not achieve uniform labeling initially, and some cells had low GFP expression even in the absence of dox. This 'leaky' GFP expression may be related to positioning of the tetO-H2B-GFP cassette in the *Col1a1* locus. Following chase, we found that CFU-F formation from the CD45⁻ GFP^int population was similar, or in some cases higher than the true GFP^hi LRCs. The GFP^int population potentially contains a mix of cells including those with dox-independent GFP expression, those who have lost some but not all label, and LRCs that did not achieve bright labeling initially. We are unable to distinguish between these populations. It is also possible that the true LRC population contains terminally differentiated quiescent cells that are present in the periosteum. Finally, CFU-F formation appears to be a feature of progenitor cells as well as stem cells, and progenitors may be enriched in the GFP^int population. The use of label retention is likely to perform better in combination with other markers that already enrich stem cell populations (*Foudi et al., 2009*; *Chakkalakal et al., 2014*). Within the αSMA-labeled population, we saw fourfold higher CFU-F frequency than αSMA-labeled non-LRCs. A previous study also reported the presence of heterogeneous LRCs in the periosteum; however, they were unable to perform functional characterization, while Nes-GFP+ periosteal cells were not LRCs (*Tournaire et al., 2020*; *Cherry et al., 2014*). Interestingly, the previous study indicated that αSMA+ cells were not LRCs; however, many antibodies to αSMA stain perivascular cells in larger blood vessels but do not detect αSMA+ cells in the cambium layer of the periosteum (*Cherry et al., 2014*). The αSMA+ cells in the cambium layer appear to be the cell subset involved in fracture healing (*Ortinau et al., 2019*; *Wang et al., 2019b*). Label retention may be a useful tool to enrich periosteal stem cells in ex vivo assays, although more robust assays for testing self-renewal such as serial transplantation are required.

One of the key findings of this study is that αSMA identifies long-term, slow-cycling, self-renewing osteoprogenitors in the adult periosteum that are functionally important for bone and cartilage formation during fracture healing. αSMA+ periosteal cells also contribute to osteoblasts in response to loading (*Matthews et al., 2020*). The lineage tracing results suggest that αSMA identifies at least two distinct osteochondroprogenitor populations in the periosteum that contribute to fracture healing. First, there is a long-term quiescent tissue-resident progenitor population that contributes to a subset of osteoblasts and chondrocytes for at least 3 months after labeling. This population is presumably re-established following fracture and contributes to healing of a second fracture in the same bone. Second, there are cells that activate αSMA after injury, as visualized in studies using real-time αSMA reporters (*Mori et al., 2016*). The majority of osteoblasts are derived from αSMA-labeled cells when tamoxifen is given at the time of fracture. The results from the fractures performed after a 2-week chase, which show slightly lower contribution than at the time of fracture but a trend toward higher contribution than after 6 or more weeks chase, suggest that there may be a third population of tissue-resident osteochondroprogenitor cells that have a limited lifespan or lack self-renewal capacity. RNAseq analysis of αSMA periosteal cells in intact bone suggested there are at least three cell populations present. While we cannot be certain which of these contains stem and progenitor cells, marker gene expression suggests that one cluster is perivascular cells, another is more fibroblast-like, and the intermediate population is likely to contain stem and progenitor cells. Unfortunately, few of the cluster marker genes were cell surface markers suitable for flow cytometry-based separation of populations.

We consistently saw lower contribution of αSMA to chondrocytes than to osteoblasts. This may be due to contribution of muscle-derived cells to chondrocytes (*Abou-Khalil et al., 2015*; *Liu et al., 2011*). Notably, αSMA expression is present in some muscle cells, both myogenic satellite cells and a separate population of stromal cells that can contribute to heterotopic bone formation (*Matthews et al., 2016*; *Kolind et al., 2015*). It is also possible that there is a periosteal population

that is restricted to chondrogenic and perhaps fibroblastic lineages, although we are not aware of any cell populations that have shown these properties in vivo. Gli1+ cells contributed to more chondrocytes than osteoblasts in a similar fracture model (*Shi et al., 2017*). Gli1+ cells also formed fibrous tissue in fractures that failed to heal due to radiation exposure, while αSMA+ cells did not (*Wang et al., 2019b*). The contribution of Gli1+ cells in the fibrous layer of the periosteum or from non-periosteal sources could explain these observations. Cxcl12-CreER also appears to identify cells that form mainly chondrocytes in a periosteal fracture callus; however, the presence of labeled cells outside the bone marrow compartment was not reported, making it difficult to identify the tissue where these cells originated (*Matsushita et al., 2020*).

Lineage tracing with Col2.3CreER indicated that there are cells the periosteum already committed to the osteogenic lineage that participate in osteoblast expansion during fracture healing. The Col2.3CreER+ cells only contribute significantly when tamoxifen is given near the time of fracture, indicating that it does not target long-lived self-renewing progenitors. Cells labeled by Prx1, Col2a1, Cathepsin K, LepR, and Mx1-driven Cre all make a major contribution to fracture callus osteoblasts; however, all these models involve either constitutive Cre expression or long-term labeling that includes differentiated cells, meaning that a combination of stem/progenitor cells and pre-osteoblasts/osteoblasts could be contributing to new osteoblast formation (*Debnath et al., 2018*; *Zhou et al., 2014b*; *Duchamp de Lageneste et al., 2018*; *Ortinau et al., 2019*; *Wang et al., 2019b*). Prx1-CreER and Sox9-CreER also label cells in the periosteum that contribute to fracture healing (*He et al., 2017*; *Xiao et al., 2020*; *Kawanami et al., 2009*). It is unclear if these populations overlap with αSMA+ cells or represent separate progenitor pools. However, our results suggest that there is heterogeneity in the cell types contributing to both osteoblasts and chondrocytes in the fracture callus, but by administering tamoxifen at the time of fracture, the majority of periosteal progenitors, particularly those that form osteoblasts, can be genetically targeted by αSMACreER.

αSMA+ cells are present in the endosteum but are less frequent than in the periosteum. αSMA is expressed in a growth-related transiently amplifying progenitor population in the trabecular region. It is also activated in cells in situations where there is a greater need for osteoblasts, such as injury, or genetic ablation of osteoblasts (*Matic et al., 2016*; *Kalajzic et al., 2008*). However, it does not appear to label cells involved in bone turnover during adulthood or consistently label long-term injury-responsive cells in the bone marrow compartment. αSMA also labels cells involved transiently in tendon growth as well as identifying progenitors of cementoblasts and periodontal fibroblasts in the periodontium (*Dyment et al., 2014*; *Roguljic et al., 2013*). αSMA+ cells are also a major contributor to healing in both these tissues as well as during reparative dentinogenesis following injury in murine molars (*Vidovic et al., 2017*). There is increasing evidence in bone tissue that different cell types could contribute to homeostasis compared to regeneration. In calvarial bones, the suture provides a niche for cells involved in both growth and healing. Axin2+ and Gli1+ cells contribute to both growth and healing; however, Prx1CreER labels a more restricted population only involved in healing (*Zhao et al., 2015*; *Maruyama et al., 2016*; *Wilk et al., 2017*). Cxcl12+ cells in the central bone marrow do not contribute to osteoblast formation under homeostatic conditions, but are activated and recruited following injury involving the bone marrow (*Matsushita et al., 2020*). αSMA+ cells in the periosteum appear to represent a population involved primarily in response to injury or increased need for bone formation.

A number of the markers and stains we evaluated did not specifically identify stem cells in the adult periosteum. CD105 was the only marker that was more prevalent in endosteum than periosteum (*Tournaire et al., 2020*) and was mostly absent from LRCs in all cell types analyzed. CD105+ periosteal cells also failed to form CFU-F. CD105 is best known as one of the markers used to identify mesenchymal stem or stromal cells in culture but has also been used as part of stains to identify or enrich progenitor populations in the periosteum and other tissues (*Chan et al., 2015*; *Debnath et al., 2018*; *Zhou et al., 2014b*; *Ortinau et al., 2019*). We demonstrated that the BCSP and SSC marker combinations, but not the pre-BSCP population, which were originally developed using neonatal bones, contain a significant proportion of mature osteoblasts when applied to adult bone tissue. This likely influences the observation that these populations expand in settings such as BMP2 treatment or fracture healing, which means some results should be revisited in future when more specific markers for adult bone-residing progenitors are developed (*Chan et al., 2015*; *Marecic et al., 2015*). Interestingly, a similar stain was used to enrich periosteal stem cells in a recent study (*Debnath et al., 2018*). Finally, CD200 was almost universally expressed on mature

osteoblasts. A number of studies suggest that CD200 is present in various stem cell populations, but it definitely does not exclude all mature cells, so should be used with caution in settings where mature osteoblasts are present (*Chan et al., 2015*; *Debnath et al., 2018*; *Lukač et al., 2020*).

In conclusion, we have demonstrated that the periosteum is highly enriched with skeletal stem and progenitor cells compared to endosteum and bone marrow. Our data demonstrate the presence of a long-term periosteum residing osteochondroprogenitor cell population that is identified by expression of αSMA, along with injury-activated αSMA+ progenitors and mature Col2.3+ osteoblasts that also contribute to the healing process. Overall, the periosteum contains quite different mesenchymal populations to those present within the bone marrow compartment, and there are multiple populations that directly contribute to new tissue formation during fracture healing.

# Materials and methods

## Key resources table

| Reagent type (species) or resource | Designation | Source or reference | Identifiers | Additional information |
|---|---|---|---|---|
| Genetic reagent (*Mus musculus*) | αSMACreER (T*g(Acta2-cre/ERT2)1Ikal*) | PMID:22083974 | MGI:5461154 | |
| Genetic reagent (*Mus musculus*) | Col2.3CreER (*B6.Cg-Tg (Col1a1-cre/ERT2)1Crm/J*) | Jackson Labs | Jax: 016241 | |
| Genetic reagent (*Mus musculus*) | Col2.3GFP (*B6.Cg-Tg(Col1a1*2.3-GFP)1Rowe/J*) | PMID:11771662 | Jax: 013134 | |
| Genetic reagent (*Mus musculus*) | Ai9 (*B6.Cg-Gt(ROSA) 26Sor$^{tm9(CAG-tdTomato)Hze}$/J*) | Jackson Labs | Jax: 007909 | |
| Genetic reagent (*Mus musculus*) | Rosa-DTA (*B6.129P2-Gt (ROSA)26Sor$^{tm1(DTA)Lky}$/J*) | Jackson Labs | Jax: 009669 | |
| Genetic reagent (*Mus musculus*) | R26-M2rtTA; TetOP-H2B-GFP (*B6;129S4-Gt(ROSA)26Sor$^{tm1(rtTA*M2)Jae}$ Col1a1$^{tm7(tetO-HIST1H2BJ/GFP)Jae}$/J*) | Jackson Labs | Jax: 016836 | |
| Genetic reagent (*Mus musculus*) | NSG (*NOD.Cg-Prkdcscid Il2rgtm1Wjl/SzJ*) | Jackson Labs | Jax: 005557 | |
| Genetic reagent (*Mus musculus*) | PdgfraCreER (*B6.129S-Pdgfra$^{tm1.1(cre/ERT2)Blh}$/J*) | Jackson Labs | Jax: 032770 | |
| Antibody | CD45 eFluor450 (rat monoclonal, clone 30-F11) | eBioscience | 48–0451 | Flow (1:400) |
| Antibody | Ter119 eFluor450 (rat monoclonal, clone TER-119) | eBioscience | 48–5921 | Flow (1:200) |
| Antibody | CD31 eFluor450 (rat monoclonal, clone 390) | eBioscience | 48–0311 | Flow (1:400) |
| Antibody | CD45 APC (rat monoclonal, clone 30-F11) | eBioscience | 17–0451 | Flow (1:400) |
| Antibody | Ter119 APC (rat monoclonal, clone TER-119) | eBioscience | 17–5921 | Flow (1:200) |
| Antibody | CD31 APC (rat monoclonal, clone 30-F11) | eBioscience | 17–0311 | Flow (1:400) |
| Antibody | CD140a APC (rat monoclonal, clone APA5) | eBioscience | 17–1401 | Flow (1:100) |
| Antibody | CD140a PE-Cy7 (rat monoclonal, clone APA5) | eBioscience | 25–1401 | Flow (1:400) |
| Antibody | CD140b APC (rat monoclonal, clone APB5) | eBioscience | 17–1402 | Flow (1:100) |
| Antibody | CD105 APC (rat monoclonal, clone MJ7/18) | eBioscience | 17–1051 | Flow (1:200) |
| Antibody | CD90.2 APC eFluor 780 (rat monoclonal, clone 53–2.1) | eBioscience | 47–0902 | Flow (1:200) |

*Continued on next page*

*Continued*

| Reagent type (species) or resource | Designation | Source or reference | Identifiers | Additional information |
|---|---|---|---|---|
| Antibody | CD90.2 BV605 (rat monoclonal, clone 30-H12) | BD Bioscience | 740334 | Flow (1:100) |
| Antibody | CD51 Biotin (rat monoclonal, clone RMV-7) | eBioscience | 13–0512 | Flow (1:100) |
| Antibody | Sca1 AlexaFluor700 (rat monoclonal, clone D7) | eBioscience | 56–5981 | Flow (1:100) |
| Antibody | Ly51 BV711 (rat monoclonal, clone 6C3) | BD Bioscience | 740691 | Flow (1:100) |
| Antibody | CD200 PE (rat monoclonal, clone OX-2) | Biolegend | 123807 | Flow (1:100) |
| Antibody | Sox9 (rabbit polyclonal) | EMD Millipore | ABE2868 | IHC (1:200) |
| Antibody | Osterix (rabbit monoclonal) | Abcam | ab209484 | IHC (1:400) |
| Antibody | PDGFRα (rabbit polyclonal) | R and D Systems | AF1062 | IHC (1:80) |
| Antibody | Goat anti rabbit AlexaFluor647 (polyclonal) | ThermoFisher | A21244 | IHC (1:300) |
| Chemical compound, drug | Streptavidin APC eFluor 780 | eBioscience | 47–4317 | Flow (1:400) |
| Chemical compound, drug | Streptavidin PE-Cy7 | eBioscience | 25–4317 | Flow (1:400) |
| Chemical compound, drug | Streptavidin APC | eBioscience | 17–4317 | Flow (1:400) |
| Chemical compound, drug | Tamoxifen | Sigma Aldrich | T5648 | 75 mg/kg i.p. |
| Chemical compound, drug | Doxycycline | Envigo | TD.01306 | 625 mg/kg in Teklad Custom Diet |
| Sequence-based reagent | Barcode oligo primer | IDT | | 5'-AAGCAGTGGTATCAACGC AGAGTACJJJJJJJJJJJJJJNNNNN NNNTTTTTTTTTTTTTTTTTTTTT TTTTTTTTTTTVN-3' |
| Sequence-based reagent | Custom primer sequence | IDT | | 5'-AATGATACGGCGACCACC GAGATCTACACGCCTGTCCG CGGAAGCAGTGGTATCAACG CAGAGT*A*C-3' |
| Sequence-based reagent | Custom read one sequence | IDT | | 5'-CGGAAGCAGTGGTATCAA CGCAGAGTAC-3' |

## Mice

All animal studies were approved by the institutional animal care and use committee at UConn Health or the University of Auckland. Mouse lines are listed in the Key Resources Table. αSMACreER and Col2.3GFP mice have been described (*Grcevic et al., 2012*; *Kalajzic et al., 2002*). R26-M2rtTA; TetOP-H2B-GFP dual homozygous mice (*Foudi et al., 2009*), Ai9 Tomato reporter mice (*Madisen et al., 2010*), Rosa-DTA (*Voehringer et al., 2008*), Col2.3CreER (*Kim et al., 2004*), and PdgfraCreER (*Chung et al., 2018*) were purchased from Jackson Labs. All these strains were maintained on a C57Bl/6J background. NSG immunodeficient mice (*NOD.Cg-Prkdcscid Il2rgtm1Wjl/SzJ*, stock # 005557) were a gift from David Rowe. All transgenic mice for experiments were bred at UConn Health and were group housed in individually ventilated cages maintained at 22 ± 2℃, 55 ± 5% humidity, and a 12 hr light-dark cycle and allowed ad libitum access to food and water. Male C57Bl/6J mice were used for experiments at the University of Auckland and were bred at the onsite facility under similar conditions. CreER was activated by tamoxifen in corn oil (75 µg/g i.p.), and two doses of tamoxifen were administered 24 hr apart unless otherwise stated. Animals were assigned to experimental groups in blocks, and cage-mates were generally exposed to the same tamoxifen regimen to avoid inadvertent exposure of low-dose tamoxifen.

## Tibia fracture

Closed tibia fractures stabilized by an intramedullary pin were generated mid-shaft as previously described (*Matthews et al., 2014*). All mice undergoing fracture lineage tracing were female and were sexually mature young adults (8–16 weeks of age at tamoxifen administration, 8–24 weeks at fracture). The presence of fracture was confirmed by X-ray (Faxitron LX 60). Animals were euthanized and limbs collected for histology 7 or 14 DPF. For secondary fractures, tibia fractures were performed and allowed to heal for 8 weeks with the pin in place. Then a secondary fracture was performed in a similar location, confirmed by X-ray, and collected 7 DPF for histological analysis.

## Histology

Bones were fixed for 5–7 days in 4% paraformaldehyde, incubated in 30% sucrose overnight and embedded in cryomatrix (Fisher Scientific). Cryosections (7 μm) were obtained using a cryostat and tape transfer system (Section-lab, Hiroshima, Japan) as previously described (*Dyment et al., 2016*). Imaging settings were kept consistent for each dataset. Fluorescent images were obtained with an Axioscan slide scanner (Zeiss). Immunostaining antibodies are listed in the Key Resources Table. For Sox9 and PDGFRα staining, sections were permeabilized in 0.1% Triton X for 15 min, blocked in 2% bovine serum albumin (BSA)/phosphate buffered saline (PBS) for 1 hr, and stained with primary antibody overnight at 4°C. For osterix staining, sections were permeabilized in 0.1% Tween-20 for 15 min, blocked in 10% goat serum, 2% BSA/PBS for 1 hr, stained with anti-Osterix antibody overnight at 4°C. Both sets of slides were washed in 0.1% Tween-20/PBS and stained with secondary antibody for 1 hr at RT, then washed thoroughly. Slides were coverslipped in 50% glycerol/PBS containing 4′,6-diamidino-2-phenylindole (DAPI). Image analysis was performed as described previously using ImageJ (*Matthews et al., 2020*). Samples were excluded from all image analysis for the following reasons: fracture was not evident or callus was absent. For osteoblast analysis, sections were excluded if they did not contain a clear fracture site. Endosteal analysis was performed only in sections that had a clear central marrow cavity present, and new bone formation evident in this region. Chondrocyte analysis was performed at day 7 only, as most cartilage is gone by day 14, and some fractures were excluded due to lack of cartilage tissue for analysis. Briefly, to evaluate Tom+ contribution to osteoblasts, regions of interest (ROIs) were drawn encompassing the fracture callus. Adjacent ROIs encompassing the central area of the callus (1 mm either side of the fracture site) as well as more distal regions were drawn (up to 6 ROIs/section), then data were pooled to determine total callus values. The DAPI channel was thresholded and separated using the watershed algorithm, then fluorescent signal for each channel was measured in nuclear regions. Standardized thresholds were set manually for each channel. We calculated the number of cells that were GFP+ or Osx+ (osteoblasts), the number that were Tom+, and the percentage of osteoblasts that were Tom+ (dual positive cells/osteoblasts). In most cases, one section/bone was analyzed. Where multiple sections were analyzed, results from different sections were consistently very similar and were pooled. A minimum of 3000 nuclei (median 9262) were evaluated for each sample. Similar analysis was performed on the bone marrow compartment when sections were suitable. For chondrocyte analysis, since there was some non-specific staining with the Sox9 antibody, only regions that demonstrated chondrocytic morphology were incorporated in the ROI analysis.

## Cell isolation

For studies comparing different tissue compartments, bone marrow, endosteum, and periosteum were isolated from hindlimbs of the same mice as previously described (*Matic et al., 2016*; *Wang et al., 2019a*). In most cases, two or three animals of the same sex were pooled to generate a sample. Briefly, after removal of muscle tissue, bone marrow was flushed and underwent red blood cell lysis (*Matthews et al., 2017*). Periosteum was scraped off the bone surface and digested in PBS containing 0.05% Collagenase P and 0.2% hyaluronidase (Sigma Aldrich, St Louis, MO, USA) for 1 hr at 37°C. To isolate endosteal cells, bones were then opened, crushed, loosely attached cells removed by washing, then digested in PBS containing 0.05% Collagenase P for 1 hr at 37°C. Cells were washed and resuspended in staining medium (SM, 2% fetal bovine serum (FBS), 1 mM ethylenediaminetetraacetic acid in PBS).

## Flow cytometry and cell sorting

Cells were stained using antibody cocktails in SM. Antibodies and secondary reagents are listed in the Key Resources Table. Analysis was performed on an LSRII or Aria II instrument (BD Biosciences), and sorting was performed on an Aria II. All experiments included unstained controls for establishing gates derived from non-fluorescent mice, and single color controls generated using periosteal or bone marrow cells. Some experiments included fluorescence minus one (FMO) controls to assist with gating. Dead cells were excluded in most experiments using DAPI staining (~50 ng/ml final concentration). Four-way sorting was performed on selected populations. Cells were sorted into 1.5 ml tubes containing 500 µl collection medium (αMEM 10% FBS). Acquisition and sorting was performed using FACSDiva software and analysis was performed in FlowJo (BD Biosciences). Experiments involved at least two biological replicates per group, and most were performed on more than one occasion to ensure consistent outcomes.

## In vitro assays

CFU-F assays were performed using freshly sorted cells. Live cell numbers were assumed to be 50% of the cells sorted by the machine. Sorted cells were resuspended in αMEM 20% FBS and seeded in six-well plates, generally at 50 cells/cm$^2$, although this varied depending on yield. In most cases, three wells were seeded per population, and at least three independent samples per population were evaluated. Cells were cultured in a 37°C humidified incubator, 5% O$_2$, 5% CO$_2$. Medium was changed on day 4, and cultures were terminated for staining on day 7–8. Cells were fixed in 10% formalin for 10 min, washed in water, then underwent crystal violet staining. For differentiation, after 7 days, medium was changed to an osteogenic/adipogenic combined cocktail containing αMEM 10% FBS, 50 µg/ml ascorbic acid, 4 mM β-glycerophosphate, 1 µM insulin, 0.5 µM rosiglitazone, and transferred to a normoxic incubator. Medium was changed after 3 days and cultures were terminated after 5 days of differentiation. Plates underwent concurrent ALP staining (86R or B5655, Sigma Aldrich) and Oil Red O staining, were imaged, then underwent crystal violet staining.

## Single cell RNAseq analysis

Periosteum was prepared as described above but with a 30-min enzymatic digestion. We sorted Tom+ cells from periosteal isolations from two to three female αSMACreER/Ai9 mice. Cells were isolated from αSMACreER/Ai9/Col2.3GFP mice at 2 days and 6 weeks after tamoxifen and from H2B-GFP/αSMACreER/Ai9 mice 13 weeks after tamoxifen and withdrawal of dox. No staining was performed and dead cells were excluded by DAPI staining. Single Tom+ cells were sorted using the index sort function on the BD Aria II directly into individual wells of a Bio-Rad hard shell 384-well plate. The plate was immediately stored in a −80°C freezer. Approximately half a plate per sample was processed for sequencing (192 cells for 2 days and 13 weeks, 216 cells for 6 weeks). Custom designed Drop-Seq barcoded primers from Integrated DNA Technologies (IDT) were delivered into each well of the 384-well plate, with primers in one well sharing the same unique cell barcode and different UMIs. An Echo 525 liquid handler was used to dispense 1 µl of primers and reaction reagents into each well for cell lysis and cDNA synthesis. Following cDNA synthesis, the contents of each well were collected and pooled into one tube using a Caliper SciClone Liquid Handler. After treatment with exonuclease I to remove unextended primers, the cDNA was polymerase chain reaction (PCR) amplified for 13 cycles. The cDNA was fragmented and amplified for sequencing with the Nextera XT DNA sample prep kit (Illumina) using custom primers that enabled the specific amplification of only the 3′ ends (see Key Resources Table). The libraries were purified, quantified, and sequenced on an Illumina MiSeq. The data preprocessing was performed using Drop-seq tools (version 1.13), with picard (version 2.17.8) for fastq to bam conversion and STAR (version 2.5.4a) for alignment to the reference genome, mm10 (version 1.2.0). Gene expression matrices were generated, and analysis was done using scanpy (version 1.3.7) (*Wolf et al., 2018*). The data is deposited in GEO under accession GSE165846. The data was filtered to remove cells with fewer than 200 genes, normalized and log transformed. The top 900 highly variable genes as measured by dispersion were used for neighborhood graph generation (nearest neighbor = 20) and dimensionality reduction with uniform manifold approximation and projection (UMAP) (*Satija et al., 2015*; *Becht et al., 2019*). Leiden community detection algorithm was used for clustering (*Traag et al.,*

*2019*). Data were visualized and plots and heat maps generated by CellView (*Bolisetty et al., 2017*).

## Label retention studies

R26-M2rtTA; TetOP-H2B-GFP (H2B-GFP) males were crossed with αSMACreER/Ai9/Ai9 females to generate experimental mice. Mice received doxin their diet (625 mg/kg) from P10 for 5 weeks. Males were removed from breeding cages prior to birth of pups to ensure mothers did not ingest dox during subsequent pregnancies. Some animals received two doses of i.p. tamoxifen at the time of dox withdrawal. The chase period was a minimum of 13 weeks. Animals that were never exposed to dox were included in all experiments as controls.

## Calvarial defect transplantation

Mice were anesthetized with ketamine/xylazine (135 and 15 mg/kg) and bilateral critical-sized circular defects (3.5 mm diameter) created in the parietal bone as previously described (*Gohil et al., 2014*). About 2500–5000 sorted αSMA-labeled cells were mixed with $2 \times 10^5$ BMSCs, resuspended in 7 µl PBS, loaded on a healos scaffold, and placed into calvaria defects of immunodeficient NSG mice (n = 6). After 5 weeks, mice were euthanized for histological analysis.

## Ablation studies

Animals were generated by crossing Rosa-DTA homozygous mice with αSMACreER, or where lineage tracing was required, crossing αSMACreER/Rosa-DTA with Ai9 and using only Cre+ animals. Fractures were performed in 7–10-week-old animals as previously described (*Novak et al., 2020*). Tamoxifen was given 0 and 1 DPF in most cases. Briefly, closed transverse diaphyseal femoral fractures were created using a blunt guillotine, after inserting a 24G needle into the intramedullary canal. To evaluate efficacy of the ablation, flow cytometry and histology were performed 4 DPF. For flow, the callus tissue was dissected and single cell suspension from individual male mice (n = 4) was generated, stained with CD45/CD31/Ter119 cocktail and DAPI. For microCT, fractured bones were collected 14 and 21 DPF, fixed in 10% formalin, and scanned (µCT40, Scanco Medical AG, Bassersdorf, Switzerland) with a voxel size of 12 µm, 55 kV, and intensity of 145 µA. We evaluated 150 slices above and below the fracture site to measure density and callus volume, and calculate bone mass. For histological analysis on day 14, frozen sections were stained using von Kossa and callus size and mineralized areas were quantified using ImageJ software as previously described (*Novak et al., 2020*). An additional group given tamoxifen 0, 2, 4, 6, and 8 DPF were included. The 21 DPF analysis group received tamoxifen 0, 2, 4 DPF. Controls with different tamoxifen regimens showed no difference between groups, so were pooled. Both male and female mice were included (day 14: Cre-: n = 14, 8M 6F; DTA D0,1: n = 10, 4M 6F; DTA D0-8: n = 8, 4M 4F).

## Statistics

All experiments include at least three biological replicates. For lineage tracing fracture experiments, we planned to include at least n = 6 per group, although long-term tracing groups in particular had more animals included to account for possible exclusions. We planned to include n = 10 for DTA studies based on our previous experience with quantitative fracture studies, but numbers and sex distribution were limited by the genotypes of available animals. Exact n values are listed in figures or figure legends, or where ranges are stated, in the source data files. Values represent the number of biological replicates (biological replicates represent one mouse/one fracture for injury studies, and one pool of two to three mice for most flow cytometry studies; most of the flow and CFU-F data involves analysis or sorts performed over multiple days or experiments). Graphs show mean ± standard error of the mean. Flow cytometry data on dot plots and in the text show mean ± standard deviation. Statistical analysis was performed in GraphPad Prism, using t-tests or one-way or two-way ANOVAs with appropriate post-hoc tests. For many of the flow cytometry datasets where different populations or tissues from one sample are evaluated, paired tests are used. Details are included in figure legends, and analysis or p values are supplied in the source data files. p<0.05 was used as the threshold for statistical significance.

## Acknowledgements

This work has been supported by Connecticut Stem Cell grant 14-SCA-UCHC-02, the Health Research Council of New Zealand Sir Charles Hercus Fellowship, and the American Society for Bone and Mineral Research Rising Star Award to BGM; Connecticut Regenerative Medicine Research Fund grant 16-RMB-UCHC-10 and NIH/NIAMS grants AR055607 and AR070813 to IK. We gratefully acknowledge the contribution of the Single Cell Biology Lab at The Jackson Laboratory for Genomic Medicine for their expert assistance with the work described in this publication. We thank Prof. David Rowe for providing NSG mice. We thank Drs Emilie Roeder and India Azevedo for assistance with fracture studies, and Dr Liping Wang for performing some of the calvarial defect surgeries.

## Additional information

### Funding

| Funder | Grant reference number | Author |
|---|---|---|
| Connecticut Innovations | 14-SCA-UCHC-02 | Brya G Matthews |
| Health Research Council of New Zealand | Sir Charles Hercus Fellowship | Brya G Matthews |
| American Society for Bone and Mineral Research | Rising Star Award | Brya G Matthews |
| National Institute of Arthritis and Musculoskeletal and Skin Diseases | AR055607 | Ivo Kalajzic |
| National Institute of Arthritis and Musculoskeletal and Skin Diseases | AR070813 | Ivo Kalajzic |
| Connecticut Innovations | 16-RMB-UCHC-10 | Ivo Kalajzic |

The funders had no role in study design, data collection and interpretation, or the decision to submit the work for publication.

### Author contributions

Brya G Matthews, Conceptualization, Formal analysis, Supervision, Funding acquisition, Validation, Investigation, Visualization, Methodology, Writing - original draft, Project administration; Sanja Novak, Formal analysis, Investigation, Methodology, Writing - review and editing; Francesca V Sbrana, Formal analysis, Investigation; Jessica L Funnell, Emma J Buckels, Formal analysis; Ye Cao, Validation, Investigation; Danka Grcevic, Investigation, Methodology, Writing - review and editing; Ivo Kalajzic, Conceptualization, Supervision, Funding acquisition, Investigation, Methodology, Writing - review and editing

### Author ORCIDs

Brya G Matthews ⓘD https://orcid.org/0000-0002-4145-4696
Sanja Novak ⓘD http://orcid.org/0000-0002-8042-932X
Ye Cao ⓘD https://orcid.org/0000-0002-1654-9060
Emma J Buckels ⓘD https://orcid.org/0000-0001-9256-5014
Ivo Kalajzic ⓘD https://orcid.org/0000-0002-6752-3442

### Ethics

Animal experimentation: The majority of the study was performed at UConn Health in an AAALAC accredited facility in accordance with the recommendations in the Guide for the Care and Use of Laboratory Animals of the National Institutes of Health. Studies were approved by the UConn Health institutional animal care and use committee (IACUC) under protocol numbers 100490-0815, 101095-0518, 101757-0221, Hz#-Dox0322e-101058 and Hz#-MCh0331e-101086. Experiments at the University of Auckland were performed in accordance with the University of Auckland Code of Ethical Conduct (CEC) and the Animal Welfare Act 1999, under Animal Ethical Committee approval 001940.

Decision letter and Author response
Decision letter https://doi.org/10.7554/eLife.58534.sa1
Author response https://doi.org/10.7554/eLife.58534.sa2

## Additional files

### Supplementary files
• Transparent reporting form

### Data availability

RNAseq data have been deposited in GEO under accession GSE165846. Source data files are provided for all figures (1–8).

The following dataset was generated:

| Author(s) | Year | Dataset title | Dataset URL | Database and Identifier |
|---|---|---|---|---|
| Matthews BG, Kalajzic I | 2021 | scRNAseq of aSMACreER/Ai9 + periosteum cells | https://www.ncbi.nlm.nih.gov/geo/query/acc.cgi?acc=GSE165846 | NCBI Gene Expression Omnibus, GSE165846 |

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
