## [Decision Letter]

**Acceptance summary:**

The major findings of the study that fracture healing is contributed by not a single but distinct stem populations, and, that periosteum contains unique progenitor population (αSMA+) that develop into osteoblasts in response to injury, are relevant in terms of identifying effective therapeutic approaches to promote healing of delayed or nonunion fractures.

**Decision letter after peer review:**

Thank you for submitting your article "Heterogeneity of murine periosteum osteochondroprogenitors involved in fracture healing" for consideration by *eLife*. Your article has been reviewed by three peer reviewers, one of whom is a member of our Board of Reviewing Editors, and the evaluation has been overseen by Clifford Rosen as the Senior Editor. The reviewers have opted to remain anonymous.

The reviewers have discussed the reviews with one another and the Reviewing Editor has drafted this decision to help you prepare a revised submission. If you have questions about the review and the concerns please contact me directly. I think the biggest issue remains novelty (see below).

As the editors have judged that your manuscript is of interest, but as described below that additional edits are required before it is published, we would like to draw your attention to changes in our revision policy that we have made in response to COVID-19 (https://elifesciences.org/articles/57162). First, because many researchers have temporarily lost access to the labs, we will give authors as much time as they need to submit revised manuscripts. We are also offering, if you choose, to post the manuscript to bioRxiv (if it is not already there) along with this decision letter and a formal designation that the manuscript is "in revision at *eLife*". Please let us know if you would like to pursue this option. (If your work is more suitable for medRxiv, you will need to post the preprint yourself, as the mechanisms for us to do so are still in development.)

All three reviewers agree that the authors have performed extensive amount of work to support the major conclusion that different osteoprogenitors in the periosteum are involved in fracture healing. However, they all have raised the same concern regarding the limited novelty of the study which is a weakness of this paper. There is now a general agreement in the literature that different cells in the periosteum contribute to the skeletal development. Since the αSMA-labeled cells characterized in this study is considered heterogeneous, the delayed healing in the α-SAM depleted mice is not surprising. It is recommended that you consider performing additional experiments based on the comments of the reviewers provided below and rewrite the manuscript focused on the novel aspects of the paper during the revision.

Reviewer #1:

Much attention has been focused recently in the skeletal biology research in characterizing the skeletal progenitors in the periosteum using various stem cell markers and evaluating their role in fracture healing. These studies have led to identification of a number of self-renewing skeletal stem cell populations (Nestin+; Mx1+αSMA+; Cathepsin K +) in the periosteum (Deveza et al., 2018; Ortinau et al., 2019; Debnath et al., 2018; Tournaire et al., 2020). Furthermore, these studies have identified stem cells in the periosteum to contribute significantly to fracture healing. In this study, Mathews et al. first characterized the putative mesenchymal stem cell markers in the periosteum, endosteum and bone marrow of adult mice. They further used lineage tracing strategy to identify αSMA cells in the periosteum to contribute to approximately 80% of osteoblasts and 40% of chondrocytes following fracture. Based on the finding that ablation of SMA+ cells reduced fracture callus volume and bone mass, it is concluded long-term, slow cycling SMA+ periosteal stem cells are functionally important for bone and cartilage formation during fracture healing. Since identifying cells that contribute to repair of fractured bone is important to the goal of developing therapies to promote healing of nonunion fractures, research in this area is of translational value. Overall, the manuscript is well written and the findings are clearly described. However, there are a number of issues that need the authors attention during the revision of this manuscript.

1) The first part of the studies is focused on characterizing the stem cell surface markers in the different regions of bone, i.e. periosteum, endosteum and bone marrow. These studies seem to clearly show evidence for presence of different populations of stem cells with different osteogenic potential in the different bone compartments. It would have been worthwhile to characterize which of these different populations, for example, Sca1+, CD51+; CD90+; Sca1-, CD51+, CD90-CD105-) are positive for SMA or other known markers of periosteal stem cells (nestin, cathepsin-k). Thus, there seems to be no connection between the first part of the studies and the subsequent studies that are focused on cells expressing SMA.

2) Fracture healing involves both intramembranous and endochondral ossification. Cartilage formation occurs early by chondrocytes which is subsequently converted to bone via endochondral ossification. Periosteal stem cells have been shown to contribute to both formation of chondrocyte-mediated endochondral bone formation as well as osteoblast-mediated intramembranous bone formation. In a recent study, Debnath et al. (Nature 562:133, 2018) showed that cathepsin-k positive periosteal cells contributed to intramembranous bone formation. In addition, recent studies also demonstrate that chondrocytes can directly transdifferentiate into bone forming osteoblasts during postnatal bone formation and fracture repair. Thus, the issues of whether there are different sub populations of stem cells in the periosteum that contribute to chondrocytes and osteoblasts during endochondral bone formation and osteoblasts during intramembranous bone formation and the extent to which the SMA+ stem cells in the periosteum contribute to the different cell types that contribute to the fracture healing remain unresolved.

3) The data in Figure 3 showing that SMA identifies long term progenitor cells in the periosteum seem very interesting. The contribution of SMA+ cells to osteoblasts was evaluated at day 7 after fracture healing. In the closed femur mid-shaft fracture model, cartilage callus is more predominant at day 7. One would, therefore, expect much more chondrocytes at day 7 fracture healing while at later stages around day 10-14, there are more osteoblasts. Accordingly, Figures 3E-G show that while there abundant SMA+ cells in the fracture callus area, the number of osteoblasts seem limited which could be due to the early time point used for histology when one would anticipate higher chondrocytes in the callus area. However, the quantitative data in Figure 3H show that 80% of Tom+ cells are osteoblasts which seem to be inconsistent with the data seen in Figure 3E.

4) To investigate the contribution of the SMA+ cells to fracture healing, the authors have used an established ROSA-DTA mouse model to ablate SMA expressing cells. The fracture phenotype in this model is not adequately characterized. The phenotype is evaluated only at one time point which is day 14 which seems early. Three and four week time points are more appropriate to evaluate if fracture healing is impaired. Time course studies of histological evaluation of cartilage and bone matrix would provide some mechanistic information on whether the reduced callus bone mass is due to reduced cartilage formation, impaired endochondral ossification or defective periosteal bridging.

Reviewer #2:

The authors investigate heterogeneity of periosteal progenitor cells using a multitude of approaches, such as markers for skeletal stem cells, label-retaining assays and lineage tracing studies of cells expressing αSMA-CreER and Col2.3-CreER. They found that SSC markers are particularly enriched in αSMA-labeled label-retaining periosteal cells that contribute to fracture healing. This strength of this study is an extensive application of cutting-edge mouse genetic and flow cytometry technologies to characterize periosteal cells.

1) The novelty of this study is not clear from the manuscript. I see the value of a wealth of important information presented here, including periosteal SSC marker expression, label-retaining cells and SMA-labeled cells in fracture healing. However, most of the major findings have been already reported elsewhere by the authors' group and others. For example, the enrichment of SSCs in the periosteum has been reported by Duchamp de Lageneste et al., 2018, and the role of αSMA+ cells in fracture healing has been reported by the authors' group (Matthews et al., 2014; Grevic et al., 2012). The way it is presented now does not really convey the progress that they made in this study. This is the critical point that the authors need to carefully address in any revision of the manuscript.

2) Heterogeneity of periosteal cells is not experimentally examined in this study, although this point is emphasized in the title. They present many findings without much cohesiveness, though each of which is quite interesting. The authors suggest that αSMA+ cells are composed of short-term and long-term cells that differentially contribute to fracture healing. Single cell RNA-seq analyses of αSMA+ cells would be essential to support this central theme.

3) The authors should present additional histological data to support their important findings. They stated that LRCs were mostly found in the inner cambium layer of the periosteum. However, no high magnification image is provided to support this claim, with appropriate counterstaining. In addition, the authors raised a very interesting theory that PaS cells might be located in the periosteum. They should perform immunostaining for PDGFRa and Sca1, and provide evidence that these cells are indeed present in the periosteum. This will enhance the novelty of this study.

Reviewer #3:

Current studies have evaluated that αSMA+ progenitor cells are long-term label retaining cells and majority of them are present in particularly on periosteum. These cells are activated with injury and become bone forming osteoblasts to contribute to the healing process. Authors provided evidences that αSMA-CreER-labeled cells express skeletal stem cell (SSC) markers and these cells maintain the expression of SSC markers for long period of time. Further, αSMA+ cells labeled at the time of injury make major contribute to bone healing process by turning into osteoblasts. However, the notion of the periosteal stem cell heterogeneity is not new and authors did not show new models, new functions and/or mechanism of αSMA+ population as compared to the authors previous study before. Further, there are many important issues to be addressed.

1) As authors have acknowledged, αSMA+ cells are heterogeneous. In fact, another group already demonstrated that a subset of αSMA-GFP+ cells overlap with Mx1+ cells and these double labeled cells are long-term repopulating stem/progenitor cells on periosteum and contribute to bone healing process with serial transplantation capability. Thus, the study with single αSMA-CreER mouse model may not further specify long-term progenitors within highly heterogenous periosteal cells and in bone healing process following the fracture.

2) LRCs in Figure 2C arrows are not convincing. There are only a few cells in periosteum but majority of GFP+ LRCs are cortical osteocytes in Figure 2C. Therefore, there are high possibility of mixture of these population in GFP high gating. Progenitor marker analysis in Figure 2H and 2I is not clear. What are the cell source for CFU-F assay from Figure 2F and 2G? periosteum? Endosteum?

3) Authors mentioned that αSMA+ cells from the bone marrow in juvenile mice became osteoblasts and osteocytes after 17 days. In Figure S3, Osteocytes are quite obvious to tell from histological section, while osteoblasts are not easy to make this conclusion without additional markers. it could be helpful to use the αSMA-CreER/Col2.3-GFP mice to investigate what proportion of αSMA+ cells turn into osteoblasts under normal bone homeostasis.

4) In previous literature, αSMA-GFP+ cells in periosteum are heterogeneous and only a subset (~25%) of them showed periosteal SSC characteristics (Ortinau et al., 2019). In Figure 3, authors claimed that αSMACreER-labeled cells are long-term osteochondroprogenitor cells. Are all αSMA-labeled cells long-term progenitors? In fact in supplemental figure, authors showed that αSMACreER-labeled cells are weak or no H2B-GFP (LRCs). Please explain this discrepancy.

5) From injury model of Figure 3, authors claimed that αSMA-labeled cells at the time of fracture make a major contribution to newly formed osteoblasts in the marrow compartment. However, it is hard to define the source of these cells as bone marrow αSMA+ cells. First, Figure S3 shows there are some αSMA+ cells in bone marrow regions (Figure S3E). Second, fracture model from current study have potential of having mixture of periosteum and bone marrow cells in both periosteum and bone marrow regions at the fracture site. Therefore, the source of those αSMA+Col2.3GFP+ cells in bone marrow compartment shown in Figure 3E is not clear.

6) In Figure 3F and H, authors showed that αSMA-labeled cells contribute to most fracture callus osteoblasts (~80%) at D+1 tam but only ~20% of osteoblasts at D+90 tam. If the labeled cells are long-term repopulating stem cells, their osteoblast contribution should be stable over the time. These results likely implicate that αSMA-labeled cells are not long-term repopulating progenitors. It will be more convincing if authors trace αSMA-labeled cells even longer time (8-10 months).

7) Figure 4G shows the transplantation of αSMA-labeled cells in a calvarial defect with a few osteoblast differentiation from αSMA+ cells. However, there is no evidence showing that there were no Tom+GFP+ cells from the beginning. Authors should include Tom+GFP+ cell transplantation as a control. Serial transplantation will be more informative.

---

## [Author Response]

All three reviewers agree that the authors have performed extensive amount of work to support the major conclusion that different osteoprogenitors in the periosteum are involved in fracture healing. However, they all have raised the same concern regarding the limited novelty of the study which is a weakness of this paper. There is now a general agreement in the literature that different cells in the periosteum contribute to the skeletal development. Since the αSMA-labeled cells characterized in this study is considered heterogeneous, the delayed healing in the αSMA depleted mice is not surprising. It is recommended that you consider performing additional experiments based on the comments of the reviewers provided below and rewrite the manuscript focused on the novel aspects of the paper during the revision.

We have revised the manuscript extensively. While the idea of heterogeneity of progenitors is becoming more commonplace, this is a recent development, and a lot of the literature, even in the past 10 years, appears to assume that there is one stem cell population that gives rise to osteoblasts in animals at various ages, and in settings including homeostasis and injury. Our study presents novel in vivo experimental data that proves there are separate populations contributing to fracture healing, and strongly suggests that the periosteum contains unique progenitor populations. We therefore believe it is an important contribution to the field.

Reviewer #1:Much attention has been focused recently in the skeletal biology research in characterizing the skeletal progenitors in the periosteum using various stem cell markers and evaluating their role in fracture healing. These studies have led to identification of a number of self-renewing skeletal stem cell populations (Nestin+; Mx1+αSMA+; Cathepsin K +) in the periosteum (Deveza et al., 2018; Ortinau et al., 2019; Debnath et al., 2018; Tournaire et al., 2020). Furthermore, these studies have identified stem cells in the periosteum to contribute significantly to fracture healing. In this study, Mathews et al. first characterized the putative mesenchymal stem cell markers in the periosteum, endosteum and bone marrow of adult mice. They further used lineage tracing strategy to identify αSMA cells in the periosteum to contribute to approximately 80% of osteoblasts and 40% of chondrocytes following fracture. Based on the finding that ablation of SMA+ cells reduced fracture callus volume and bone mass, it is concluded long-term, slow cycling SMA+ periosteal stem cells are functionally important for bone and cartilage formation during fracture healing. Since identifying cells that contribute to repair of fractured bone is important to the goal of developing therapies to promote healing of nonunion fractures, research in this area is of translational value. Overall, the manuscript is well written and the findings are clearly described. However, there are a number of issues that need the authors attention during the revision of this manuscript.1) The first part of the studies is focused on characterizing the stem cell surface markers in the different regions of bone, i.e. periosteum, endosteum and bone marrow. These studies seem to clearly show evidence for presence of different populations of stem cells with different osteogenic potential in the different bone compartments. It would have been worthwhile to characterize which of these different populations, for example, Sca1+, CD51+; CD90+; Sca1-, CD51+, CD90-CD105-) are positive for SMA or other known markers of periosteal stem cells (nestin, cathepsin-k). Thus, there seems to be no connection between the first part of the studies and the subsequent studies that are focused on cells expressing SMA.

We have reorganized the presentation of the data to improve the flow and the integration of the αSMA data with the rest of the results focusing on characterizing periosteal progenitor populations. We have added quantification of αSMA+ cells within the Sca1/CD51 populations, and CD90/CD105 populations in Figure 2—figure supplement 2D, Results subsection “Putative mesenchymal stem cell markers are enriched in periosteum”. Since Nestin and CatK are not cell surface markers that can be evaluated without transgenic lines, we were not able to assess the overlap of these populations. We have, however, included all these markers, and various others used in recent studies, in the evaluation of αSMA+ cells’ gene expression (Figure 3D). CatK expression was detected in 79/529 αSMA+ cells. Most of these cells (90%) were within the progenitor and fibroblast-like clusters (#1 and 2). Nestin was detected in 15/529 cells, the majority (67%) of which were in the perivascular-like cluster 3.

2) Fracture healing involves both intramembranous and endochondral ossification. Cartilage formation occurs early by chondrocytes which is subsequently converted to bone via endochondral ossification. Periosteal stem cells have been shown to contribute to both formation of chondrocyte-mediated endochondral bone formation as well as osteoblast-mediated intramembranous bone formation. In a recent study, Debnath et al. (Nature 562:133, 2018) showed that cathepsin-k positive periosteal cells contributed to intramembranous bone formation. In addition, recent studies also demonstrate that chondrocytes can directly transdifferentiate into bone forming osteoblasts during postnatal bone formation and fracture repair. Thus, the issues of whether there are different sub populations of stem cells in the periosteum that contribute to chondrocytes and osteoblasts during endochondral bone formation and osteoblasts during intramembranous bone formation and the extent to which the SMA+ stem cells in the periosteum contribute to the different cell types that contribute to the fracture healing remain unresolved.

We have performed a much more thorough characterization and quantification of αSMA+ cell contribution to fracture healing than other studies in the literature that have characterized other progenitor markers that we are aware of. While Debnath et al. report that CatK^+^ progenitors show intramembranous bone formation upon transplantation, they demonstrate some contribution to fibrocartilage following fracture in vivo (see extended data Figure 8 in their publication) (Debnath et al., 2018). The CatK model is problematic for lineage tracing since it is a constitutive Cre that doesn’t enable temporal understanding of the CatK expression. Reaching a consensus on mesenchymal/osteoprogenitor cell identity will require the research community to compile data from numerous lineage tracing models to confirm contribution of different cell populations. There is a great deal more to understand, but we believe that our data using αSMACreER and Col2.3CreER clearly demonstrate that there are different populations contributing to healing, and some of these populations are lineage-restricted. We therefore think this is an important contribution to the field that can be considered in the context of other results.

3) The data in Figure 3 showing that SMA identifies long term progenitor cells in the periosteum seem very interesting. The contribution of SMA+ cells to osteoblasts was evaluated at day 7 after fracture healing. In the closed femur mid-shaft fracture model, cartilage callus is more predominant at day 7. One would, therefore, expect much more chondrocytes at day 7 fracture healing while at later stages around day 10-14, there are more osteoblasts. Accordingly, Figures 3E-G show that while there abundant SMA+ cells in the fracture callus area, the number of osteoblasts seem limited which could be due to the early time point used for histology when one would anticipate higher chondrocytes in the callus area. However, the quantitative data in Figure 3H show that 80% of Tom+ cells are osteoblasts which seem to be inconsistent with the data seen in Figure 3E.

The data the reviewer is referring to (now Figure 1E-I) actually present the percent of Col2.3GFP+ osteoblasts that are Tom+ (dual positive cells/GFP+) rather than the proportion of SMA cells differentiating into osteoblasts. The reviewer’s comment about the dynamics of osteoblast and chondrocyte numbers in the callus are correct – there are generally fewer osteoblasts at day 7, and almost no chondrocytes by day 14, which is why this cell type was only analyzed in the day 7 callus. We analyzed both time points for osteoblasts as we hypothesized that there could be different cells contributing to osteoblasts at different phases of healing. Our data do not support this hypothesis, and we restricted certain analyses such as the 90 day chase to day 7 only. The different time points are now mentioned in the Results section: “The contribution of αSMA-labeled cells was similar in both the initial intramembranous bone formation in the day 7 callus, and the more fully mineralized day 14 callus.” We did not present additional data about the fracture composition as it is well-established, and we felt it made it more difficult to clearly present our findings. However, we clarified the parameter that is presented in both the Materials and methods: *“*We calculated the number of cells that were GFP+ or Osx+ (osteoblasts), the number that were Tom+, and the percentage of osteoblasts that were Tom+ (dual positive cells/osteoblasts).”; and the figure legend for Figure 1. In addition, we have added much more detailed source data for Figure 1G-I which shows all the key outputs from the image analysis. By comparing the GFP+ values in column N for day 7 and day 14, it is clear that the proportion of osteoblasts in the callus tends to increase over the time course. We hope this will make it clearer to readers what analysis is presented.

4) To investigate the contribution of the SMA+ cells to fracture healing, the authors have used an established ROSA-DTA mouse model to ablate SMA expressing cells. The fracture phenotype in this model is not adequately characterized. The phenotype is evaluated only at one time point which is day 14 which seems early. Three and four week time points are more appropriate to evaluate if fracture healing is impaired. Time course studies of histological evaluation of cartilage and bone matrix would provide some mechanistic information on whether the reduced callus bone mass is due to reduced cartilage formation, impaired endochondral ossification or defective periosteal bridging.

We performed the DTA ablation studies as a proof of principle that αSMA+ cells have a functional role in the fracture healing process, but limited the evaluation as this is not a physiologically relevant model, and there are potentially other effects of ablating αSMA+ cells that affect the health of the animals. We have added additional data to what is now Figure 7, and the accompanying Results section. We have confirmed that the callus area and mineralized area are reduced histologically at 14 DPF, and now present microCT data from 21 DPF. The phenotype at day 21 is more subtle, but still indicates a deficiency in fracture healing. The following statement has been added to the results to describe this process: “Overall, our data demonstrate that partial ablation of αSMA+ progenitors reduced callus formation and delayed mineralization. The less severe phenotype at 21 DPF suggests this may be a delay in healing rather than complete disruption of the process, although the fractures from mice with αSMA-ablation probably never reach the size of control calluses. Healing in the DTA mice may still progress due to compensation of αSMA+ progenitors that escaped ablation, or larger contribution from αSMA-negative progenitors in this setting.”

Reviewer #2:The authors investigate heterogeneity of periosteal progenitor cells using a multitude of approaches, such as markers for skeletal stem cells, label-retaining assays and lineage tracing studies of cells expressing αSMA-CreER and Col2.3-CreER. They found that SSC markers are particularly enriched in αSMA-labeled label-retaining periosteal cells that contribute to fracture healing. This strength of this study is an extensive application of cutting-edge mouse genetic and flow cytometry technologies to characterize periosteal cells.1) The novelty of this study is not clear from the manuscript. I see the value of a wealth of important information presented here, including periosteal SSC marker expression, label-retaining cells and SMA-labeled cells in fracture healing. However, most of the major findings have been already reported elsewhere by the authors' group and others. For example, the enrichment of SSCs in the periosteum has been reported by Duchamp de Lageneste et al., 2018, and the role of αSMA+ cells in fracture healing has been reported by the authors' group (Matthews et al., 2014; Grevic et al., 2012). The way it is presented now does not really convey the progress that they made in this study. This is the critical point that the authors need to carefully address in any revision of the manuscript.

We have reorganized the manuscript in a way that better highlights the important findings and improves the cohesiveness. While some of the observations have been reported in other studies, we believe that it is important to have multiple lines of evidence from a variety of sources to support the presence and definition of different populations of stem/progenitor cells within the periosteum. More importantly, while other studies imply heterogeneity of healing-related progenitors, our study shows direct evidence of this through in vivo lineage tracing of different populations.

2) Heterogeneity of periosteal cells is not experimentally examined in this study, although this point is emphasized in the title. They present many findings without much cohesiveness, though each of which is quite interesting. The authors suggest that αSMA+ cells are composed of short-term and long-term cells that differentially contribute to fracture healing. Single cell RNA-seq analyses of αSMA+ cells would be essential to support this central theme.

Flow cytometry data indicates heterogeneity with respect to surface marker expression, however we have extended this by adding single cell RNAseq data for αSMA+ cells to the revised manuscript (Figure 3 and supplements). This data has indicated heterogeneity in the population, and shifts in relative abundance of the populations over time. Further studies will be required, potentially with the use of new genetic tools, to confirm the roles of these subpopulations, but we believe this new data strengthens our findings.

3) The authors should present additional histological data to support their important findings. They stated that LRCs were mostly found in the inner cambium layer of the periosteum. However, no high magnification image is provided to support this claim, with appropriate counterstaining. In addition, the authors raised a very interesting theory that PaS cells might be located in the periosteum. They should perform immunostaining for PDGFRa and Sca1, and provide evidence that these cells are indeed present in the periosteum. This will enhance the novelty of this study.

High magnification of H2B-GFP expression in the periosteum is shown in what is now Figure 4—figure supplement 1. We have added additional annotation to this figure to indicate labeled cells within the inner layer of the periosteum. Unfortunately, we don’t have counterstained images available, but we believe that the DAPI staining clearly demonstrates the location within the periosteum.

We have added additional data addressing the use of PDGFRα, and this is presented in Figure 8 and the accompanying Results section. Both immunostaining and Cre-mediated labeling indicated that PDGFRα expression is very prevalent in the periosteum, likely more so than our flow data suggested. Unfortunately we were unable to achieve dual Sca1/PDGFRα staining of suitable quality for publication in the current circumstances. However, the new data we present indicates serious caveats of using PDGFRα as an osteoprogenitor marker, primarily due to clear expression in the osteoblast layer, that we thought were important to include, and we have altered the Results and Discussion accordingly.

Reviewer #3:Current studies have evaluated that αSMA+ progenitor cells are long-term label retaining cells and majority of them are present in particularly on periosteum. These cells are activated with injury and become bone forming osteoblasts to contribute to the healing process. Authors provided evidences that αSMA-CreER-labeled cells express skeletal stem cell (SSC) markers and these cells maintain the expression of SSC markers for long period of time. Further, αSMA+ cells labeled at the time of injury make major contribute to bone healing process by turning into osteoblasts. However, the notion of the periosteal stem cell heterogeneity is not new and authors did not show new models, new functions and/or mechanism of αSMA+ population as compared to the authors previous study before. Further, there are many important issues to be addressed.1) As authors have acknowledged, αSMA+ cells are heterogeneous. In fact, another group already demonstrated that a subset of αSMA-GFP+ cells overlap with Mx1+ cells and these double labeled cells are long-term repopulating stem/progenitor cells on periosteum and contribute to bone healing process with serial transplantation capability. Thus, the study with single αSMA-CreER mouse model may not further specify long-term progenitors within highly heterogenous periosteal cells and in bone healing process following the fracture.

We have taken advantage of the inducible Cre model that allows us to characterize cells ex vivo and track their potential in vivo in a physiological setting. The data from other studies including the Ortinau study, and Debnath et al. 2018 (Debnath et al., 2018; Ortinau et al., 2019), as well as our data, suggest that transplantation results do not completely recapitulate in vivo differentiation potential as we generally see osteogenic differentiation only in following transplantation, while observing osteochondrogenic potential in the fracture setting. In the revised manuscript, we have changed the order to emphasize novel findings, added new data to confirm self-renewal, as well as single cell RNAseq data to further characterize the αSMA+ population.

2) LRCs in Figure 2C arrows are not convincing. There are only a few cells in periosteum but majority of GFP+ LRCs are cortical osteocytes in Figure 2C. Therefore, there are high possibility of mixture of these population in GFP high gating. Progenitor marker analysis in Figure 2H and 2I is not clear. What are the cell source for CFU-F assay from Figure 2F and 2G? periosteum? Endosteum?

We have annotated the H2B-GFP histology more clearly in what is now Figure 4—figure supplement 1.

We are confident that our cell preparation method for the periosteum does not result in osteocyte contamination as we scrape the periosteum surface and do not digest cortical bone. Our endosteal preparation involves digestion of bone pieces, however we have previously demonstrated using Dmp1CreER/Ai9 mice, which show reporter expression only in osteocytes in the absence of tamoxifen treatment (Tam-), that we get almost no Tom+ cells in these preparations (see Matic, Matthews et al. 2016, Supplemental Figure S6C)

The following statement has been added to the Results: “Our periosteal isolation procedure excludes bone tissue and therefore osteocytes, and we have previously demonstrated that our endosteal preparations contain almost no osteocytes.”

We have added additional annotation to the figure (now Figure 4), to clarify that the CFU-F data is from periosteal cells, and the populations analyzed in 4F and G are spelt out more clearly in the figure legend.

3) Authors mentioned that αSMA+ cells from the bone marrow in juvenile mice became osteoblasts and osteocytes after 17 days. In Figure S3, Osteocytes are quite obvious to tell from histological section, while osteoblasts are not easy to make this conclusion without additional markers. it could be helpful to use the αSMA-CreER/Col2.3-GFP mice to investigate what proportion of αSMA+ cells turn into osteoblasts under normal bone homeostasis.

This data is replication of data we published in 2012 (Grcevic et al., 2012) that we included for completeness, and as we had not previously published the longer term tracing data. We have therefore not completed this study as our data clearly indicate contribution to osteoblasts and osteocytes, and it is outside the major scope of this study.

4) In previous literature, αSMA-GFP+ cells in periosteum are heterogeneous and only a subset (~25%) of them showed periosteal SSC characteristics (Ortinau et al., 2019). In Figure 3, authors claimed that αSMACreER-labeled cells are long-term osteochondroprogenitor cells. Are all αSMA-labeled cells long-term progenitors? In fact in supplemental Figure, authors showed that αSMACreER-labeled cells are weak or no H2B-GFP (LRCs). Please explain this discrepancy.

We have noted at various points of the manuscript that αSMA-labeled cells are heterogeneous, and the fracture lineage tracing data (now Figure 1) suggests that only a subset have long-term progenitor characteristics. This is now more clearly demonstrated using single cell RNAseq data presented in the new Figure 3. As mentioned in the results, over 20% of αSMA-labeled cells were LRCs, and we have added additional annotation to what is now Figure 4—figure supplement 1 to show this histologically as well.

5) From injury model of Figure 3, authors claimed that αSMA-labeled cells at the time of fracture make a major contribution to newly formed osteoblasts in the marrow compartment. However, it is hard to define the source of these cells as bone marrow αSMA+ cells. First, Figure S3 shows there are some αSMA+ cells in bone marrow regions (Figure S3E). Second, fracture model from current study have potential of having mixture of periosteum and bone marrow cells in both periosteum and bone marrow regions at the fracture site. Therefore, the source of those αSMA+Col2.3GFP+ cells in bone marrow compartment shown in Figure 3E is not clear.

It is true that we cannot completely exclude some bone marrow contribution in the fracture model, however the weight of evidence in the literature indicates that when present, the periosteum is the major contributor to fracture callus formation. We have also performed preliminary studies with a periosteal scratch model where the cortex is not injured, and demonstrated formation of a callus containing αSMA-labeled cells (data not shown). In addition, there is clear evidence indicating that bone marrow/endosteum-derived cells do not form fibrocartilage in vivo, and we see fibrocartilage near the fracture site in most day 7 samples (Colnot et al., 2009). On this basis, combined with our data indicating αSMA+ cells are present but much rarer in the endosteum (new Figure 2D), we have persisted with our interpretation that the cells contributing to the fracture callus osteoblasts and chondrocytes are primarily derived from the periosteum. The majority of the bone formation in the bone marrow compartment is around the pin and is usually in the proximal area of the bone at least 1mm from the fracture site. We also never see evidence of chondrocytes. On this basis, we are fairly confident that bone marrow-resident cells are responsible for this bone formation. The text has been modified to clarify this:

“Following the pin insertion and fracture, bone formation also occurs inside the marrow compartment, particularly in the proximal diaphysis surrounding the pin over 1mm from the fracture site (Figure 1—figure supplement 3).”

6) In Figure 3F and H, authors showed that αSMA-labeled cells contribute to most fracture callus osteoblasts (~80%) at D+1 tam but only ~20% of osteoblasts at D+90 tam. If the labeled cells are long-term repopulating stem cells, their osteoblast contribution should be stable over the time. These results likely implicate that αSMA-labeled cells are not long-term repopulating progenitors. It will be more convincing if authors trace αSMA-labeled cells even longer time (8-10 months).

We agree with the reviewer than it would be interesting to do even longer tracing, but given the amount of time this study would take, it is outside the scope of revisions for this journal. As explained in the manuscript, the difference between the labeling at the time of fracture, and 90 days before is likely to be primarily due to cells that activate αSMA expression soon after injury. It is also notable that contribution stabilizes between 42 and 90 days chase.

7) Figure 4G shows the transplantation of αSMA-labeled cells in a calvarial defect with a few osteoblast differentiation from αSMA+ cells. However, there is no evidence showing that there were no Tom+GFP+ cells from the beginning. Authors should include Tom+GFP+ cell transplantation as a control. Serial transplantation will be more informative.

Almost all the αSMA-Tom+ cells isolated for transplantation were GFP-. This is shown in the flow plots from the cell sorting in Author response image 1. On this basis, it is not feasible to sort and transplant Tom+GFP+ cells as it is such a rare population. In other studies, Col2.3GFP+ cells show minimal growth following sorting, at least in vitro (Siclari et al., 2013).

**Author response image 1. sa2fig1:** Dot plot of periosteum sample used for sorting αSMA-labeled cells for calvarial defect transplantation. Note that almost all Tom+ cells are GFP-.

We agree with the reviewer that self-renewal was not adequately assessed. We have included a different experiment to address self-renewal of these cells in the new Figure 6 where two cycles of fractures were performed (tamoxifen labeling, initial fracture, healing for 8 weeks, secondary fracture at same site). This shows significant contributions of initially αSMA-labeled cells to secondary fracture healing, even if the initial tamoxifen labeling was 14 days before fracture. This strongly suggests identification of at least a subset of self-renewing progenitors. We believe this experimental design is more physiologically relevant, particularly as there are many technical challenges involved in serial transplantation experiments.